# PESSIMISTIC MODEL SELECTION FOR OFFLINE DEEP REINFORCEMENT LEARNING

## ABSTRACT

Deep Reinforcement Learning (DRL) has demonstrated great potentials in solving sequential decision making problems in many applications. Despite its promising performance, practical gaps exist when deploying DRL in real-world scenarios. One main barrier is the over-fitting issue that leads to poor generalizability of the policy learned by DRL. In particular, for offline DRL with observational data, model selection is a challenging task as there is no ground truth available for performance demonstration, in contrast with the online setting with simulated environments. In this work, we propose a pessimistic model selection (PMS) approach for offline DRL with a theoretical guarantee, which features a provably effective framework for finding the best policy among a set of candidate models. Two refined approaches are also proposed to address the potential bias of DRL model in identifying the optimal policy. Numerical studies demonstrated the superior performance of our approach over existing methods.

## 1 INTRODUCTION

The success of deep reinforcement learning (Mnih et al., 2013; Henderson et al., 2018) (DRL) often leverages upon executive training data with considerable efforts to select effective neural architectures. Deploying online simulation to learn useful representations for DRL is not always realistic and possible especially in some high-stake environments, such as automatic navigation (Kahn et al., 2018; Hase et al., 2020), dialogue learning (Jaques et al., 2020), and clinical applications (Tang et al., 2020a). *Offline reinforcement learning* (Singh & Sutton, 1996; Levine et al., 2020; Agarwal et al., 2020) (OffRL) has prompted strong interests (Paine et al., 2020; Kidambi et al., 2020) to empower DRL toward training tasks associated with severe potential cost and risks. The idea of OffRL is to train DRL models with only logged data and recorded trajectories. However, with given observational data, designing a successful neural architecture in OffRL is often expensive (Levine et al., 2020), requiring intensive experiments, time, and computing resources.

Unlike most aforementioned applications with online interaction, *Offline* tasks for reinforcement learning often suffer the challenges from insufficient observational data from offline collection to construct a universal approximated model for fully capturing the temporal dynamics. Therefore, relatively few attempts in the literature have been presented for provide a pipeline to automate the development process for model selection and neural architecture search in OffRL settings. Here, model selection refers to selecting the best model (e.g., the policy learned by a trained neural network) among a set of candidate models (e.g. different neural network hyperparameters).

In this work, we propose a novel model selection approach to automate OffRL development process, which provides an evaluation mechanism to identify a well-performed DRL model given offline data. Our method utilizes statistical inference to provide uncertainty quantification on value functions trained by different DRL models, based on which a pessimistic idea is incorporated to select the best model/policy. In addition, two refined approaches are further proposed to address the possible biases of DRL models in identifying the optimal policy. In this work, we mainly focus on deep Q-network (Mnih et al., 2013; 2015) (DQN) based architectures, while our proposed methods can be extended to other settings. Figure 1 demonstrates the superior performance of the proposed pessimistic model selection (PMS) method in identifying the best model among 70 DRL models of different algorithms on one navigation task (See Appendix C for details), compared with the model selection method by (Tang & Wiens, 2021) which uses three offline policy evaluation (OPE)

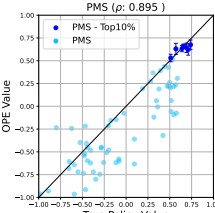 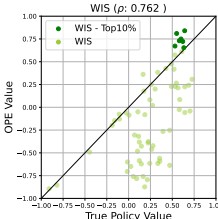 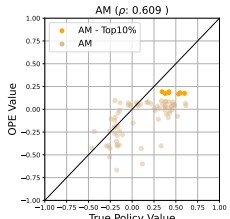 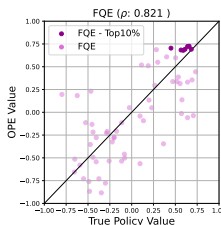

**Figure 1:** Model selection algorithms for offline DQN learning: (a) proposed pessimistic model selection (PMS); (b) weighted importance sampling (WIS) (Gottesman et al., 2018); (c) approximate model (AM) (Voloshin et al., 2019); (d) fitted Q evaluation (FQE) (Le et al., 2019). In this figure, the algorithms are trained and evaluated in a navigation task ($\mathbf{E}_2$) discussed in Section 7 and Appendix C.

estimates for validation. Specifically, based on the derived confidence interval of the OPE value for each candidate model, the final selected model by our PMS method is the one that has the largest lower confidence limit, which exactly has the largest true OPE value among all candidate models. In contrast, none of three OPE estimates used for model selection by Tang & Wiens (2021) can identify the best model due to the inevitable overfitting issue during the validation procedure.

To close this section, we summarize the contributions of this work as follows:

- We propose a novel PMS framework, which targets finding the best policy from given candidate models (e.g., neural architecture, hyperparameters, etc) with offline data for DQN learning. Unlike many existing methods, our approach essentially does not involve additional hyperparameter tuning except for two interpretable parameters.

- Leveraging asymptotic analysis in statistical inference, we provide uncertainty quantification on each candidate model, based on which our method can guarantee that the worst performance of finally selected model is the best among all candidate models. See Corollary 1 for more details.

- To address potential biases of candidate models in identifying the optimal policy, two refined approaches are proposed, one of which can be shown to have regret bounded by the smallest error bound among all candidate models under some technical conditions (See Corollary 2). To the best of our knowledge, this is the first model-selection method in offline DRL with such a guarantee.

- The numerical results demonstrate that the proposed PMS shows superior performance in different DQN benchmark environments.

## 2 RELATED WORK

**Model Selection for Reinforcement Learning:** Model selection has been studied in online decision-making environments (Fard & Pineau, 2010; Lee & Taylor, 2014). Searching nearly optimal online model is a critical topic for online bandits problems with limited information feed-backs. For linear contextual bandits, Abbasi-Yadkori et al. (2011); Chu et al. (2011) are aiming to find the best worst-case bound when the optimal model class is given. For model-based reinforcement learning, Pacchiano et al. (2020) introduces advantages of using noise augmented Markov Decision Processes (MDP) to archive a competitive regret bound to select an individual model with constraints for ensemble training. Recently, Lee et al. (2021) utilized an online algorithm to select a low-complexity model based on a statistical test. However, most of the previous model selection approaches are focused on the online reinforcement learning setting. Very few works including Farahmand & Szepesvári (2011); Paine et al. (2020); Su et al. (2020); Yang et al. (2020); Kuzborskij et al. (2021); Tang & Wiens (2021); Xie & Jiang (2021) are focused on the offline setting. In particular, (Su et al., 2020; Yang et al., 2020; Kuzborskij et al., 2021) focus on model selection for OPE problem. (Farahmand & Szepesvári, 2011; Xie & Jiang, 2021) select the best model/policy based on minimizing the Bellman error, while the first approach requires an additional tuning and latter does not. (Paine et al., 2020; Tang & Wiens, 2021) proposed several criteria to perform model selection in OffRL and mainly focused on numerical studies. In this work, we provide one of the first model selection approaches based on statistical inference for RL tasks with offline data collection.

**Offline-Policy Learning:** Training a DRL agent with offline data collection often relies on batch-wise optimization. Batch-Constrained deep Q-learning (Fujimoto et al., 2019) (BCQ) is considered one OffRL benchmark that uses a generative model to minimize the distance of selected actions to the batch-wise data with a perturbation model to maximize its value function. Other popular OffRL

approaches, such as behavior regularized actor-critic (BRAC) (Wu et al., 2019), and random ensemble mixture (Agarwal et al., 2020) (REM) (as an optimistic perspective on large dataset), have also been studied in RL Unplugged (RLU) (Gulcehre et al., 2020) benchmark together with behavior cloning (Bain & Sammut, 1995; Ross & Bagnell, 2010) (BC), DQN, and DQN with quantile regression (Dabney et al., 2018) (QR-DQN). RLU suggests a naive approach based on human experience for offline policy selection, which requires independent modification with shared domain expertise (e.g., Atari environments) for tuning each baseline. Meanwhile, how to design a model selection algorithm for OffRL remains an open question. Motivated by the benefits and the challenges as mentioned earlier of the model selection for *offline* DRL, we aim to develop a unified approach for model selection in offline DRL with theoretical guarantee and interpretable tuning parameters.

## 3    BACKGROUND AND NOTATIONS

Consider a time-homogeneous Markov decision process (MDP) characterized by a tuple $\mathcal{M} = (\mathcal{S}, \mathcal{A}, p, r, \gamma)$, where $\mathcal{S}$ is the state space, $\mathcal{A}$ is the action space, $p$ is the transition kernel, i.e., $p(s'|s, a)$ is the probability mass (density) of transiting to $s'$ given current state-action $(s, a)$, $r$ is the reward function, i.e., $\mathbb{E}(R_t|S_t = s, A_t = a) = r(s, a)$ for $t \geq 0$, and $0 \leq \gamma < 1$ is a discount factor. For simplifying presentation, we assume $\mathcal{A}$ and $\mathcal{S}$ are both finite. But our method can also be applied in continuous cases. Under this MDP setting, it is sufficient to consider stationary Markovian policies for optimizing discounted sum of rewards (Puterman, 1994). Denote $\pi$ as a stationary Markovian policy mapping from the state space $\mathcal{S}$ into a probability distribution over the action space. For example, $\pi(a|s)$ denotes the probability of choosing action $a$ given the state value $s$. One essential goal of RL is to learn an optimal policy that maximizes the value function. Define $V^\pi(s) = \sum_{t=0}^{+\infty} \gamma^t \mathbb{E}^\pi[R_t|S_0 = s]$ and then the optimal policy is defined as $\pi^* \in \text{argmax}_\pi \{\mathcal{V}(\pi) \triangleq (1 - \gamma) \sum_{s \in \mathcal{S}} V^\pi(s)\nu(s)\}$, where $\nu$ denotes some reference distribution function over $\mathcal{S}$. In addition, we denote Q-function as $Q^\pi(s, a) = \sum_{t=0}^{+\infty} \gamma^t \mathbb{E}^\pi(R_t|A_0 = a, S_0 = s)$ for $s \in \mathcal{S}$ and $a \in \mathcal{A}$. In this work, we consider the OffRL setting. The observed data consist of $N$ trajectories, corresponding to $N$ independent and identically distributed copies of $\{(S_t, A_t, R_t)\}_{t \geq 0}$. For any $i \in \{1, \cdots, n\}$, data collected from the $i$th trajectory can be summarized by $\{(S_{i,t}, A_{i,t}, R_{i,t}, S_{i,t+1})\}_{0 \leq t < T}$, where $T$ denotes the termination time. We assume that the data are generated by some fixed stationary policy denoted by $b$.

Among many RL algorithms, we focus on Q-learning type of methods. The foundation is the optimal Bellman equation given below.

$$Q^*(s, a) = \mathbb{E}[R_t + \gamma \max_{a' \in \mathcal{A}} Q^*(S_{t+1}, a') \,|\, S_t = s, A_t = a], \tag{1}$$

where $Q^*$ is called optimal Q-function, i.e., Q-function under $\pi^*$. Among others, fitted q-iteration (FQI) is one of the most popular RL algorithms (Ernst et al., 2005). FQI leverages supervised learning techniques to iteratively solve the optimal Bellman equation (1) and shows competitive performance in OffRL.

To facilitate our model-selection algorithm, we introduce the discounted visitation probability, motivated by the marginal importance sampling estimator in (Liu et al., 2018). For any $t \geq 0$, let $p_t^\pi(s, a)$ denote the $t$-step visitation probability $\text{Pr}^\pi(S_t = s, A_t = a)$ assuming the actions are selected according to $\pi$ at time $1, \cdots, t$. We define the discounted visitation probability function as $d^\pi(s, a) = (1 - \gamma) \sum_{t \geq 0} \gamma^t p_t^\pi(s, a)$. To adjust the distribution from behavior policy to any target policy $\pi$, we use the discounted probability ratio function defined as

$$\omega^{\pi,\nu}(s, a) = \frac{d^\pi(s)\pi(a|s)}{\frac{1}{T}\sum_{t=0}^{T-1} p_t^b(s, a)}, \tag{2}$$

where $p_t^b(s, a)$ is the $t$-step visitation probability under the behavior policy $b$, i.e., $\text{Pr}^b(S_t = s, A_t = a)$. The ratio function $\omega^{\pi,\nu}(s, a)$ is always assumed well defined. The estimation of ratio function is motivated by the observation that for every measurable function $f$ defined over $\mathcal{S} \times \mathcal{A}$,

$$\mathbb{E}[\frac{1}{T}\sum_{t=0}^{T-1} \omega^{\pi,\nu}(S_t, A_t)(f(S_t, A_t) - \gamma \sum_{a' \in \mathcal{A}} \pi(a' \,|\, S_{t+1})f(S_{t+1}, a'))]$$
$$= (1 - \gamma)\mathbb{E}_{S_0 \sim \nu}[\sum_{a \in \mathcal{A}} \pi(a \,|\, S_0)f(a, S_0)], \tag{3}$$

based on which several min-max estimation methods has been proposed such as (Liu et al., 2018; Nachum et al., 2019; Uehara & Jiang, 2019); We refer to (Uehara & Jiang, 2019, Lemma 1) for a formal proof of equation (3).

Finally, because our proposed model selection algorithm relies on an efficient evaluation of any target policy using batch data, we introduce three types of offline policy evaluation estimators in the existing RL literature. The first type is called direct method via estimating Q-function, based on the relationship that $\mathcal{V}(\pi) = (1 - \gamma) \sum_{s \in \mathcal{S}, a \in \mathcal{A}} \pi(a|s)Q(s, a)\nu(s)$. The second type is motivated by the importance sampling (Precup, 2000). Based on the definition of ratio function, we can see $\mathcal{V}(\pi) = \mathbb{E}[\frac{1}{T}\sum_{t=0}^{T-1} \omega^{\pi,\nu}(S_t, A_t)R_t]$, from which a plugin estimator can be constructed. The last type of OPE methods combines the first two types of methods and construct a so-called doubly robust estimator (Kallus & Uehara, 2019; Tang et al., 2020b). This estimator is motivated by the efficient influence function of $\mathcal{V}(\pi)$ under a transition-sampling setting and the model that consists of the set of all observed data distributions given by arbitrarily varying the initial, transition, reward, and behavior policy distributions, subject to certain minimal regularity and identifiability conditions (Kallus & Uehara, 2019), i.e.,

$$\frac{1}{T}\sum_{t=0}^{T-1} \omega^{\pi,\nu}(S_t, A_t)(R_t + \gamma \sum_{a \in \mathcal{A}} \pi(a|S_{t+1})Q^{\pi}(S_{t+1}, a) - Q^{\pi}(S_t, A_t))$$
$$+ (1-\gamma)\mathbb{E}_{S_0 \sim \nu}[\sum_{a \in \mathcal{A}} \pi(a|S_0)Q^{\pi}(S_0, a)] - \mathcal{V}(\pi). \tag{4}$$

A nice property of doubly robust estimators is that as long as either the Q-function $Q^{\pi}(s, a)$ or the ratio function $\omega^{\pi,\nu}(s, a)$ can be consistently estimated, the final estimator of $\mathcal{V}(\pi)$ is consistent (Robins et al., 1994; Jiang & Li, 2015; Kallus & Uehara, 2019; Tang et al., 2020b). Furthermore, a doubly robust estimator based on (4) can achieve semiparametric efficiency under the conditions proposed by (Kallus & Uehara, 2019), even if nuisance parameters are estimated via black box models such as deep neural networks. Therefore such an estimator is particularly suitable under the framework of DRL. Our proposed algorithm will rely on this doubly robust type of OPE estimator.

## 4 PESSIMISTIC MODEL SELECTION (PMS) FOR BEST POLICY

In this section, we discuss our pessimistic model selection approach. For the ease of presentation, we focus on the framework of (deep) Q-learning, where policy optimization is performed via estimating the optimal Q-function. While this covers a wide range of state-of-the-art RL algorithms such as FQI (Ernst et al., 2005), DQN (Mnih et al., 2013) and QR-DQN (Dabney et al., 2018), we remark that our method is not restricted to this class of algorithms.

Suppose we have total number of $L$ candidate models for policy optimization, where each candidate model will output an estimated policy, say $\hat{\pi}_l$ for $1 \leq l \leq L$. Our goal is to select the best policy among $L$ policies during our training procedure. Note that these $L$ models can be different deep neural network architectures, hyper-parameters, and various classes of functions for approximating the optimal Q-function or policy class, etc.

### 4.1 DIFFICULTIES AND CHALLENGES

Given a candidate $l$ among $L$ models, we can apply for example FQI using the whole batch data $\mathcal{D}_n$ to learn an estimate of $Q^*$ as $\widehat{Q}_l$ and an estimated optimal policy $\hat{\pi}_l$ defined as $\hat{\pi}_l(a|s) \in \text{argmax}_{a \in \mathcal{A}}\widehat{Q}_l(s, a)$, for every $s \in \mathcal{S}$. In order to select the final policy, one may use a naive greedy approach to choose some $\tilde{l}$ such that $\tilde{l} \in \text{argmax}_l \mathbb{E}_{S_0 \sim \nu}[\sum_{a \in \mathcal{A}} \hat{\pi}_l(a|s)\widehat{Q}_l(S_0, a)]$, as our goal is to maximize $\mathcal{V}(\pi)$. However, using this criterion will lead to over-fitting. Specifically, due to the distributional mismatch between the behavior policy and target policies, which is regarded as a fundamental challenge in OffRL (Levine et al., 2020), we may easily overestimate Q-function, especially when some state-action pairs are not sufficiently visited in the batch data. This issue becomes more serious when we apply max-operator during our policy optimization procedure. Such observations have already been noticed in recent works, such as (Kumar et al., 2019; 2020; Paine et al., 2020; Yu et al., 2020; Tang & Wiens, 2021; Jin et al., 2021). Therefore, it may be inappropriate to use this criterion for selecting the best policy among $L$ models.

One may also use cross-validation procedure to address the issue of over-fitting or overestimating Q-function for model selection. For example, one can use OPE approaches on the validate dataset to evaluate the performance of estimated policies from the training data set (see Tang & Wiens (2021) for more details). However, since there is no ground truth for the value function of any policies, the OPE procedure on the validation dataset cannot avoid involving additional tuning on hyperparameters. Therefore such a procedure may still incur a large variability due to the over-fitting issue. In addition, arbitrarily splitting the dataset for cross-validation and ignoring the Markov dependent structure will cause additional errors, which should be seriously taken care of.

## 4.2 SEQUENTIAL MODEL SELECTION

In the following, we propose a pessimistic model selection algorithm for finding an optimal policy among $L$ candidate models. Our goal is to develop an approach to estimate the value function under each candidate model during our policy optimization procedure with theoretical guarantee. The proposed algorithm is motivated by recent development in statistical inference of sequential decision making (Luedtke & Van Der Laan, 2016; Shi et al., 2020). The idea is to first estimate optimal Q-function $Q^*$, optimal policy $\pi^*$ and the resulting ratio function based on a chunk of data, and evaluate the performance of the estimated policy on the next chunk of data using previously estimated nuisance functions. Then we combine the first two chunks of data, perform the same estimation procedure and evaluation on the next chunk of data. The framework of MDP provides a nature way of splitting the data.

Specifically, denote the index of our batch dataset $\mathcal{D}_n$ as $J_0 = \{(i, t) : 1 \leq i \leq n, 0 \leq t < T\}$. We divide $J_0$ into $O$ number of non-overlapping subsets, denoted by $J_1, \cdots, J_O$ and the corresponding data subsets are denoted by $\mathcal{D}_1, \cdots, \mathcal{D}_O$. Without loss of generality, we assume these data subsets have equal size. We require that for any $1 \leq o_1 < o_2 \leq O$, any $(i_1, t_1) \in J_{o_1}$ and $(i_2, t_2) \in J_{o_2}$, either $i_2 \neq i_1$ or $t_1 < t_2$. For $1 \leq o \leq O$, denote the aggregate chunks of data as $\bar{\mathcal{D}}_o = \left\{ (S_{i,t}, A_{i,t}, R_{i,t}, S_{i,t+1}), (i, t) \in \bar{J}_o = J_1 \cup \cdots \cup J_o \right\}$.

We focus on FQI algorithm for illustrative purpose and it should be noticed that our algorithm can be applied to other RL algorithms. Starting from the first chunk of our batch data, at the $o$-th step ($o = 1, \cdots, O - 1$), for each candidate model $l = 1, \cdots, L$, we apply FQI on $\bar{\mathcal{D}}_o$ to compute $\widehat{Q}_l^{(o)}$ as an estimate of optimal Q-function and obtain $\hat{\pi}_l^{(o)}$ correspondingly such that $\hat{\pi}_l^{(o)}(a|s) \in \text{argmax}_{s \in \mathcal{A}} \widehat{Q}_l^{(o)}(s, a)$ for every $s \in \mathcal{S}$. Additionally, we compute an estimate of ratio function $\omega^{\hat{\pi}_l^{(o)}, \nu}$ using $\bar{\mathcal{D}}_o$ by many existing algorithms such as Nachum et al. (2019). Denote the resulting estimator as $\widehat{\omega}^{\hat{\pi}_l^{(o)}, \nu}$. The purpose of estimating this ratio function is to improve the efficiency and robustness of our value function estimation for each candidate model. Then we compute the estimated value function of $\hat{\pi}_l^{(o)}$ on $\mathcal{D}_{o+1}$ as

$$\hat{\mathcal{V}}_{\mathcal{D}_{o+1}}(\hat{\pi}_l^{(o)}) = (1 - \gamma)\mathbb{E}_{S_0 \sim \nu}[\sum_{a_0 \in \mathcal{A}} \hat{\pi}_l^{(o)}(a_0|S_0)\widehat{Q}_l^{(o)}(S_0, a_0)] \tag{5}$$

$$+ \mathbb{E}_{\mathcal{D}_{o+1}}[\widehat{\omega}^{\hat{\pi}_l^{(o)}, \nu}(S, A)(R + \gamma \sum_{a' \in \mathcal{A}} \hat{\pi}_l^{(o)}(a'|S')\widehat{Q}_l^{(o)}(S', a') - \widehat{Q}_l^{(o)}(S, A))], \tag{6}$$

where $\mathbb{E}_{\mathcal{D}_{o+1}}$ denotes the empirical average over the $(o + 1)$ chunk of dataset and $(S, A, R, S')$ is one transition tuple in $\mathcal{D}_{o+1}$. While one can aggregate $\hat{\mathcal{V}}_{\mathcal{D}_{o+1}}(\hat{\pi}_l^{(o)})$ for $1 \leq o \leq (O - 1)$ to evaluate the performance of $L$ models, the uncertainty of these estimates due to the finite sample estimation should not be ignored. Therefore, in the following, we derive an uncertainty quantification of our estimated value function for each candidate model, for performing model selection. Based on equation (4), (conditioning on $\bar{\mathcal{D}}_o$), the variance of $\hat{\mathcal{V}}_{\mathcal{D}_{o+1}}(\hat{\pi}_l^{(o)})$ is

$$\sigma^2(\hat{\pi}_l^{(o)}) = \mathbb{E}[\{\widehat{\omega}^{\hat{\pi}_l^{(o)}, \nu}(S, A)(R + \gamma \sum_{a' \in \mathcal{A}} \hat{\pi}_l^{(o)}(a'|S')\widehat{Q}_l^{(o)}(S', a') - \widehat{Q}_l^{(o)}(S, A))\}^2], \tag{7}$$

where $(S, A, S')$ is a transition tuple with $(S, A)$ follows some stationary distribution. See Assumption 1. Correspondingly we have an estimate defined as

$$\hat{\sigma}_{o+1}^2(\hat{\pi}_l^{(o)}) = \mathbb{E}_{\mathcal{D}_{o+1}}[\{\widehat{\omega}^{\hat{\pi}_l^{(o)}, \nu}(S, A)(R + \gamma \sum_{a' \in \mathcal{A}} \hat{\pi}_l^{(o)}(a'|S')\widehat{Q}_l^{*(o)}(S', a') - \widehat{Q}_l^{*(o)}(S, A))\}^2]. \tag{8}$$

The estimation procedure stops once we have used all our offline data and denote the final estimated policy as $\hat{\pi}_l$ for each $l = 1, \cdots, L$. Notice that $\hat{\pi}_l = \hat{\pi}_l^{(O)}$. Finally, we compute the weighted average of all the intermediate value functions as our final evaluation of the estimated policy $\hat{\pi}_l$, i.e.,

$$\hat{\mathcal{V}}(\hat{\pi}_l) = \left(\sum_{o=1}^{O-1} \frac{1}{\hat{\sigma}_{o+1}(\hat{\pi}_l^{(o)})}\right)^{-1} \left(\sum_{o=1}^{O-1} \frac{\hat{\mathcal{V}}_{\mathcal{D}_{o+1}}(\hat{\pi}_l^{(o)})}{\hat{\sigma}_{o+1}(\hat{\pi}_l^{(o)})}\right). \tag{9}$$

In Section 5, we show that under some technical conditions, the following asymptotic result holds:

$$\frac{\sqrt{nT(O-1)/O}\left(\hat{\mathcal{V}}(\hat{\pi}_l) - \mathcal{V}(\hat{\pi}_l)\right)}{\hat{\sigma}(l)} \Longrightarrow \mathcal{N}(0,1), \tag{10}$$

where $\hat{\sigma}(l) = (O-1)(\sum_{o=1}^{O-1}\{\sigma_{o+1}(\hat{\pi}_l^{(o)})\}^{-1})^{-1}$, $\Longrightarrow$ refers to weak convergence when either $n$ or $T$ goes to infinity, and $\mathcal{N}(0,1)$ refers to the standard normal distribution. Based on the asymptotic result in (10), we can construct a confidence interval for the value function of each policy $\hat{\pi}_l$. Given a confidence level $\alpha$, for each $l$, we can compute $U(l) = \hat{\mathcal{V}}(\hat{\pi}_l) - z_{\alpha/2}\sqrt{O/nT(O-1)}\hat{\sigma}(l)$, where $z_{\alpha/2}$ is $(1 - \frac{\alpha}{2})$-quantile of the standard normal distribution. Our final selected one is $\hat{l} \in \text{argmax}_{1 \leq l \leq L} U(l)$.

The use of $U(l)$ is motivated by the recent proposed pessimistic idea to address the overestimation issue of value (or Q) function in the OffRL setting. See Kumar et al. (2019; 2020); Jin et al. (2021); Xie et al. (2021); Uehara & Sun (2021); Zanette et al. (2021) for details. The final output of our algorithm is $\hat{\pi}_{\hat{l}}$ and an outline of the proposed algorithm can be found in Algorithm 1. As we can see, our algorithm is nearly tuning free, which provides great flexibility in real-world applications. The only two adjustable parameters is $O$ and $\alpha$. The size of $O$ balances the computational cost and the finite-sample accuracy of evaluating each candidate model. In specific, we can indeed show that the variance of the estimated value function by our algorithm can achieve the semi-parametric efficiency bound, which is best one can hope for. So in the asymptotic sense, the effect of $O$ is negligible. In the finite-sample setting, we believe the performance will be discounted by a factor $\sqrt{O-1/O}$. Therefore, if $O$ is large enough, $\sqrt{O-1/O}$ will have a mere effect on the performance. See Theorem 1. However, using large $O$ will result in a large computational cost. As a sacrifice for the nearly tuning free algorithm, we need to apply OffRL algorithms $O$ times for each candidate model. The parameter $\alpha$ determines how worst the performance of each policy we should use to evaluate each policy. See Corollary 1 for more insights.

---

**Algorithm 1:** Pessimistic Model Selection (PMS) for OffRL

**Input:** Dataset $\mathcal{D}_n$ and $L$ candidate models for estimating optimal Q-function and policy; We divide $\mathcal{D}_n$ into non-overlapping subsets denoted by $\mathcal{D}_1, \cdots, \mathcal{D}_O$. We require that for any $1 \leq o_1 < o_2 \leq O$, any $(i_1, t_1) \in J_{o_1}$ and $(i_2, t_2) \in J_{o_2}$, either $i_2 \neq i_1$ or $t_1 \leq t_2$.

1 **for** $l \in L$ **do**
2      **for** $o = 1$ ***to*** $O - 1$ **do**
3          For $l \in L$ models, construct the optimal $\widehat{Q}_l^{(o)}$ and $\hat{\pi}_l^{(o)}$ using $\bar{\mathcal{D}}_o$ data subset.
4          Compute $\widehat{\omega}^{\hat{\pi}_l^{(o)}, \nu}$ using $\bar{\mathcal{D}}_o$ by Nachum et al. (2019) and min-max solver for (3).
5          Compute $\hat{\mathcal{V}}_{\mathcal{D}_{o+1}}(\hat{\pi}_l^{(o)})$ and $\hat{\sigma}_{o+1}^2(l)$ using $\mathcal{D}_{o+1}$ given in (5) and (8) respectively.
6      For $l$-th model, we compute $U(l) = \hat{\mathcal{V}}(\hat{\pi}_l) - z_{\alpha/2}\sqrt{nT(O-1)/O}\hat{\sigma}(l)$, where $\hat{\mathcal{V}}(\hat{\pi}_l)$ and $\hat{\sigma}(l)$ are given in (9) and (10) respectively.
7 Pick $\hat{l} = \arg\max_l U(l)$ as the selected model and run the algorithm on full dataset to obtain $\hat{\pi}_{\hat{l}}$.
8 **Return** $\hat{\pi}_{\hat{l}}$.

---

## 5 THEORETICAL RESULTS

In this section, we justify our asymptotic result given in (10). We use $O_p$ to denote the stochastic boundedness. Before that, we make several technical assumptions:

**Assumption 1** *The stochastic process $\{A_t, S_t\}_{t \geq 0}$ is stationary with stationary distribution $p_\infty$.*
**Assumption 2** *For every $1 \leq l \leq L$ and $1 \leq o \leq O$, we have $\mathbb{E}|\mathcal{V}(\hat{\pi}_l^{(o)}) - \mathcal{V}(\pi^*)| \leq C_0(nT/O)^{-\kappa}$, for some constant $C_0$ and $\kappa > 1/2$.*

Assumption 1 is standard in the existing literature such as (Kallus & Uehara, 2019). Assumption 2 is key to our developed asymptotic results developed. This assumption essentially states that all candidate models are good enough so that eventually their value functions will converge to that of the optimal one. This implies that there is no asymptotic bias in identifying the optimal policy. While this is reasonable thanks to the capability of deep neutral networks, which has demonstrated their empirical success in many RL applications, such an assumption could still be strong. In Section 6, we try to relax this assumption and provide two remedies for addressing possibly biased estimated policies. In addition, Assumption 1 also requires that the convergence rates of value functions under estimated policies are fast enough. This has been shown to hold under the margin condition on $\pi^*$, see e.g., (Hu et al., 2021) for more details.

**Assumption 3** *For every* $1 \leq l \leq L$ *and* $1 \leq o \leq O - 1$, *suppose* $\mathbb{E}_{(S,A)\sim p_\infty} |\widehat{Q}_l^{(o)}(S, A) - Q^{\hat{\pi}_l^{(o)}}(S, A)|^2 = O_p\{(nT/O)^{-2\kappa_1}\}$ *for some constant* $\kappa_1 \geq 0$. *In addition,* $\widehat{Q}_l^{(o)}$ *is uniformly bounded almost surely.*

**Assumption 4** *For every* $1 \leq l \leq L$ *and* $1 \leq o \leq O - 1$, *suppose* $\mathbb{E}_{(S,A)\sim p_\infty} |\widehat{\omega}^{\hat{\pi}_l^{(o)},\nu}(S, A) - \omega^{\hat{\pi}_l^{(o)},\nu}(S, A)|^2 = O_p\{(nT/O)^{-2\kappa_2}\}$ *for some constant* $\kappa_2 \geq 0$. *In addition, both* $\omega^{\hat{\pi}_l^{(o)},\nu}$ *and* $\widehat{\omega}^{\hat{\pi}_l^{(o)},\nu}$ *are uniformly bounded above and below away from* $0$ *almost surely.*

**Assumption 5** *For every* $1 \leq l \leq L$ *and* $1 \leq o \leq O - 1$, $\sigma^2(\hat{\pi}_l^{(o)})$ *and* $\hat{\sigma}_{o+1}^2(\hat{\pi}_l^{(o)})$ *are bounded above and below from* $0$ *almost surely.*

Assumptions 3 and 4 impose high-level conditions on two nuisance functions. Our theoretical results only require $\kappa_1 + \kappa_2 > 1/2$, which is a mild assumption. For example, if considered parametric models for both Q-function and ratio function, then $\kappa_1 = \kappa_2 = 1/2$. If considered nonparametric models for these two nuisance functions such as deep neural networks, then $1/4 < \kappa_1, \kappa_2 < 1/2$ can be obtained under some regularity conditions. See Fan et al. (2020) and Liao et al. (2020); Uehara et al. (2021) for the convergence rates of Q-function and ratio function by non-parametric models respectively. In addition, Assumption 5 is a mild assumption, mainly for theoretical justification. Then we have the following main theorem as a foundation of our proposed algorithm.

**Theorem 1** *Under Assumptions 1-5, we have*

$$(\sqrt{nT(O-1)/O}\left(\hat{\mathcal{V}}(\hat{\pi}_l) - \mathcal{V}(\hat{\pi}_l)\right))/\hat{\sigma}(l) \Longrightarrow \mathcal{N}(0,1). \tag{11}$$

Theorem 1 provides an uncertainty quantification of each candidate model used in policy optimization. Such uncertainty quantification is essential in OffRL as data are often limited. We highlight the importance of such results in Appendix A. A consequent result following Theorem 1 validates the proposed Algorithm 1:

**Corollary 1** $\liminf\limits_{nT \to \infty} \Pr(\mathcal{V}(\hat{\pi}_{\hat{l}}) \geq \max_{1 \leq l \leq L} \mathcal{V}(\hat{\pi}_l) - 2z_{\alpha/2}\sqrt{nT(O-1)/O}\hat{\sigma}(l)) \geq 1 - L\alpha$ *under Assumptions 1-5.*

As can be seen clearly from Corollary 1 and the proposed PMS, with a large probability (by letting $\alpha$ small), we consider the worst performance of each candidate model $\hat{\pi}_l$ in the sense of the lower confidence limit of the value function, and then select the best one among all models.

## 6 TWO REFINED APPROACHES

In this section, we relax Assumption 2 by allowing possibly non-negligible bias in estimating the optimal policy and introduce two refined approaches for addressing this issue. Instead of imposing Assumption 2, we make an alternative assumption below.

**Assumption 6** *For every* $1 \leq l \leq L$, *there exists* $B(l)$ *such that* $\max_{1 \leq o \leq (O-1)} |\mathcal{V}(\hat{\pi}_l^{(o)}) - \mathcal{V}(\pi^*)| \leq B(l)$ *almost surely.*

Assumption 6 is a very mild assumption. It essentially states that the biases for all our intermediate value function estimates are bounded by some constant, which is much weaker than Assumption 2. In this case, the asymptotic results in (11) may not hold in general. Correspondingly, we have the following result.

**Theorem 2** *Under Assumptions 1, 3-6, for every* $1 \leq l \leq L$, *the following inequality holds.*

$$\liminf\limits_{nT \to \infty} \Pr\left(|\mathcal{V}(\pi^*) - \hat{\mathcal{V}}(\hat{\pi}_l)| \leq z_{\alpha/2}\sqrt{O/nT(O-1)}\hat{\sigma}(l) + B(l)\right) \geq 1 - \alpha. \tag{12}$$

Motivated by Lepski's principle (Lepski & Spokoiny, 1997) from nonparametric statistics and (Su et al., 2020) studying the model selection of OPE, we consider the following refined model-selection procedure to find the best policy. We first rank $L$ candidate models in an non-increasing order based on the value of $\hat{\sigma}(l)$, i.e., for $1 \le i < j \le L$, $\hat{\sigma}(i) \ge \hat{\sigma}(j)$. Then for $i$-th model, we construct an interval as $I(l) = [\hat{\mathcal{V}}(\hat{\pi}_l) - 2z_{\alpha/(2L)}\sqrt{O/nT(O-1)}\hat{\sigma}(l), \hat{\mathcal{V}}(\hat{\pi}_l) + 2z_{\alpha/(2L)}\sqrt{O/nT(O-1)}\hat{\sigma}(l)]$. Finally the optimal model/policy we choose is $\hat{\pi}_{\hat{i}}$ such that $\hat{i} = \max\{i : 1 \le i \le L, \cap_{1 \le j \le i} I(j) \ne \emptyset\}$. To show this procedure is valid, we need to make one additional assumption.

**Assumption 7** *There exists a $\zeta < 1$ such that for $1 \le i \le L$, $B(i) \le B(i+1)$ and $\zeta\hat{\sigma}(i) \le \hat{\sigma}(i+1) \le \hat{\sigma}(i)$ almost surely.*

This assumption is borrowed from Su et al. (2020). It typically assumes that after sorting our model based on $\hat{\sigma}(l)$, the bias of estimated policy is monotonically increasing and the standard deviation is monotonically deceasing but not too quickly. This is commonly seen when all candidate estimators exhibit some bias-variance trade-off phenomena. Define the following event

$$\mathcal{E} = |\hat{\mathcal{V}}(\hat{\pi}_{\hat{i}}) - \mathcal{V}(\pi^*)| \le 6(1 + \zeta^{-1}) \min_{1 \le i \le L}\{B(i) + z_{\alpha/(2L)}\sqrt{O/nT(O-1)}\hat{\sigma}(i)\}. \quad (13)$$

Then we have the following theoretical guarantee for our refined procedure.

**Corollary 2** *Under Assumptions 1, 3-7, we have $\liminf_{nT \to \infty} \Pr(\mathcal{E}) \ge 1 - \alpha$. If we further assume that for any $\delta > 0$, with probability at least $1 - \delta$, for every $1 \le i \le L$, $|\mathcal{V}(\hat{\pi}_i) - \hat{\mathcal{V}}(\hat{\pi}_i)| \le c(\delta)\log(L)\hat{\sigma}(i)/\sqrt{NT}$ for some constant $c(\delta)$, then $\liminf_{nT \to \infty} \Pr(\bar{\mathcal{E}}) \ge 1 - \alpha - \delta$, where*

$$\bar{\mathcal{E}} = |\mathcal{V}(\hat{\pi}_{\hat{i}}) - \mathcal{V}(\pi^*)| \le 3(1 + \zeta^{-1}) \min_{1 \le i \le L}\{B(i) + (c(\delta)\log(L) + z_{\alpha/(2L)})\sqrt{O/nT(O-1)}\hat{\sigma}(i)\}. \quad (14)$$

The additional assumption (i.e., the high probability bound) in Corollary 2 can be shown to hold by the empirical process theory under some technical conditions (Van de Geer, 2000). Hence Corollary 2 provides a strong guarantee that the regret of the final selected policy is bounded by the smallest error bound among all $L$ candidate policies. Note that Assumption 3 imposed here could be strong.

**Another refined approach:** Notice that the above refined approach indeed focuses on OPE estimates to select the best policy with regret warranty. The rough motivation behind is to find a policy that has the smallest estimation error to the optimal one. However, such procedure may not directly match the goal of maximizing the value function in OffRL. To relieve this issue , we can alternatively choose the final policy as $\hat{\pi}_{\hat{\hat{i}}}$ such that $\hat{\hat{i}} = \text{argmax}_{1 \le i \le \hat{i}} \hat{\mathcal{V}}(\hat{\pi}_i) - 2z_{\alpha/2}\sqrt{nT(O-1)/O}\hat{\sigma}(i)$, where the argmax is taken over $\hat{i}$ models. This approach can be viewed as a combination of PMS and the above refined approach. By adopting this approach, candidate models with large biases are firtly removed by the truncation on $\hat{i}$. Then, we use the idea of PMS to select the best model having the best worst performance among the remaining candidates. Unfortunately, we do not have theoretical guarantee for this combined approach.

## 7 EXPERIMENTAL RESULTS

We select six DQN environments ($\mathbf{E}_1$ to $\mathbf{E}_6$) from open-source benchmarks (Brockman et al., 2016; Juliani et al., 2018) to conduct numerical experiments, as shown in Fig. 5 of Appendix C. These tasks of deployed environments cover different domains that include tabular learning (Fig 5(a)); automatic navigation in a geometry environment with a physical ray-tracker (Fig 5(b)); Atari digital gaming (Fig 5(c) and (d)), and continuous control (Fig 5(e) and (f)). We provide detailed task description and targeted reward for each environment in Appendix C.

**Experiment setups.** To evaluate the performance of PMS with DQN models in offRL, we choose different neural network architectures under five competitive DRL algorithms including DQN by (Mnih et al., 2013; 2015), BCQ by (Fujimoto et al., 2019), BC by (Bain & Sammut, 1995; Ross & Bagnell, 2010), BRAC by (Wu et al., 2019) from RLU benchmarks, and REM by (Agarwal et al., 2020). Within each architecture, 70 candidate models are created by assigning different hyperparameters and training setups. See Appendix C.1 for details. We then conduct performance evaluation of different OffRL model selection methods on these generated candidate models.

**Evaluation procedure.** We utilize validation scores from OPE for each model selection algorithm, which picks the best (or a good) policy from the candidate set of size $L$ based on its own criterion.

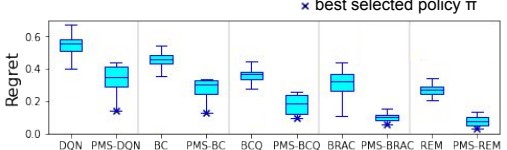

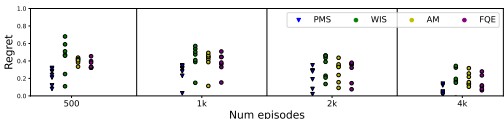

**Figure 2:** Box plots of model selection performance from offline learning in each DRL algorithms for $\mathbf{E}_2$.

**Figure 3:** Sensitivity analysis for different training data size. PMS attains the best performance and has the least sensitivity.

Regret is used as the evaluation metric for each candidate. The regret for model $l$ is defined as $\mathcal{V}(\pi_{l*}) - \mathcal{V}(\hat{\pi}_l)$, where $l^* = \arg\max_{l'=1...L} \mathcal{V}(\pi_{l'})$ corresponds to the candidate policy with the best OPE validation performance. In our implementation, we treat $\pi_{l*}$ as $\pi^*$, the oracle but unknown best possible policy. A small regret is desirable after model selection. Note the optimal regret is not zero since we can only use data to obtain $\hat{\pi}_l$ instead of $\pi_l$ for each model. We provide additional top-k regret and precision results from Figure 6 to 16 in Appendix C.

**Performance comparison.** As highlighted in Fig. 1 in the introduction, we report estimated OPE values by different model selection approaches, i.e. PMS and three methods by (Tang & Wiens, 2021), versus the true OPE values. In this experiment, we consider 70 DQN models under the above mentioned five DRL algorithms, i.e., 14 models are considered for each architecture. We use fewer models for each DRL algorithm mainly for clear presentation. By using the confidence interval constructed by our PMS procedure, our method is able to correctly select the top models, while the other three methods fail. To further investigate the performance of PMS, we implement model selection among 70 models within each DRL algorithm separately. Fig. 2 shows the box plots of averaged regret over six environments after OPE per neural network architecture. Each subfigure contains results from one particular DRL algorithm with different hyperparameters or training setups. The left box plot refers to the regrets of all 70 models and the right one represents the regrets of top $10\%$ models selected by the proposed PMS method. Note that the right box plot is a subset of the left one. The results show that our proposed PMS successfully helps to select models with the best policies and improve the average regret by a significant margin. In particular, PMS-REM-based models attain the lowest regrets, due to the benefit from its ensemble process. Detailed results for each environment is given in Appendix C.

**Sensitivity analysis.** Fig. 3 compares different selection algorithms with varying training data size. PMS outperforms others across all scales, and larger number of episodes gives smaller variation and lower sensitivity.

**PMS algorithm with refinements.** We replicate our experiments on in the offline navigation task in $\mathbf{E}_2$ (*Banana Collector*) for 30 times and report regrets of top $10\%$ models selected by PMS and two refinements in Fig. 4. As we can see, while the overall performances of the proposed three model selection methods are sim-

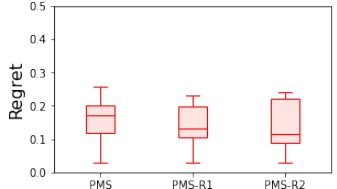

**Figure 4:** PMS and its refinements (R1/R2).

ilar, two refined approaches have better regrets than PMS in terms of median, demonstrating their promising performances in identifying the best model. OPE results have been also evaluated also in DRL tasks with $\mathbf{E}_1$ and $\mathbf{E}_3$ to $\mathbf{E}_6$, where the refinement algorithms (PMS R1/R2) have only a small relative $\pm$ 0.423 % performance difference compared to its original PMS setups.

# 8 CONCLUSION

We propose a new theory-driven model selection framework (PMS) for offline deep reinforcement learning based on statistical inference. The proposed pessimistic mechanism is warrants that the worst performance of the selected model is the best among all candidate models. Two refined approaches are further proposed to address the biases of DRL models. Extensive experimental results on six DQN environments with various network architectures and training hyperparameters demonstrate that our proposed PMS method consistently yields improved model selection performance over existing baselines. The results suggest the effectiveness of PMS as a powerful tool toward automating model selection in offline DRL.

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

# A  COMMENTS ON ASYMPTOTIC RESULTS

We remark here that all theoretical justification in this paper is based on asymptotics. It might be possible to investigate finite sample regimes when one has an exact confidence interval instead of (11) or a non-asymptotic bound. However, having an exact confidence interval might require some model specification of the value function, and using non-asymptotic bounds might require additional tuning steps (e.g., constants in many concentration inequalities), which is beyond the scope of this paper. In addition, as seen from our empirical evaluations below, with a relatively large sample size, the proposed model selection approach performs well.

# B  TECHNICAL PROOFS

*Notations*: The notation $\xi(N) \lesssim \theta(N)$ (resp. $\xi(N) \gtrsim \theta(N)$) means that there exists a sufficiently large (resp. small) constant $c_1 > 0$ (resp. $c_2 > 0$) such that $\xi(N) \leq c_1\theta(N)$ (resp. $\xi(N) \geq c_2\theta(N)$) for some sequences $\theta(N)$ and $\xi(N)$ related to $N$. In the following proofs, $N$ often refers to some quantity related to $n$ and $T$.

**Lemma 1 and its proof** : Let $J$ denotes some index of our batch data $\mathcal{D}_n$. Define

$$\phi(J, Q^\pi, \omega^{\pi,\nu}, \pi) = \frac{1}{|J|} \sum_{(i,t)\in J} \omega^{\pi,\nu}(S_{i,t}, A_t) \left( R_{i,t} + \gamma \sum_{a'\in\mathcal{A}} \pi(a'|S_{i,t+1})Q^\pi(S_{i,t+1}, a') - Q^\pi(S_{i,t}, A_{i,t}) \right),$$

where $|J|$ is the cardinality of the index set $J$, e.g., $|J_o| = \frac{nT}{O}$ for every $1 \leq o \leq O$. Then we have the following Lemma 1 as an intermediate result to Theorem 1.

**Lemma 1** *Under Assumptions 1 and 3-5, for every $1 \leq l \leq L$ and $1 \leq o \leq O - 1$, the following asymptotic equivalence holds.*

$$\sqrt{\frac{nT}{O}} \left\{ \hat{\mathcal{V}}_{\mathcal{D}_{o+1}}(\hat{\pi}_l^{(o)}) - \mathcal{V}(\hat{\pi}_l^{(o)}) \right\} = \sqrt{\frac{nT}{O}} \phi(J, Q^{\hat{\pi}^{*(o)}}, \omega^{\hat{\pi}_l^{(o)},\nu}, \hat{\pi}_l^{(o)}) + o_p(1), \qquad (15)$$

*where $o_p(1)$ refers to a quantity that converges to 0 as $n$ or $T$ goes to infinity.*

The proof is similar to that of Theorem 7 in Kallus & Uehara (2019). First, notice that

$$\sqrt{\frac{nT}{O}} \left\{ \hat{\mathcal{V}}_{\mathcal{D}_{o+1}}(\hat{\pi}_l^{(o)}) - \mathcal{V}(\hat{\pi}_l^{(o)}) \right\}$$

$$= \sqrt{\frac{nT}{O}} \left\{ \phi(J, \widehat{Q}^{\hat{\pi}^{*(o)}}, \widehat{\omega}^{\hat{\pi}_l^{(o)},\nu}, \hat{\pi}_l^{(o)}) - \phi(J, Q^{\hat{\pi}^{*(o)}}, \omega^{\hat{\pi}_l^{(o)},\nu}, \hat{\pi}_l^{(o)}) \right.$$

$$\left. + (1-\gamma)\mathbb{E}_{S_0\sim\nu}[\sum_{a\in\mathcal{A}} \hat{\pi}_l^{(o)}(a|S_0)Q^{\hat{\pi}_l^{(o)}}(S_0, a)] - (1-\gamma)\mathbb{E}_{S_0\sim\nu}[\sum_{a\in\mathcal{A}} \hat{\pi}_l^{(o)}(a|S_0)Q^{\hat{\pi}_l^{(o)}}(S_0, a)] \right\}$$

$$+ \sqrt{\frac{nT}{O}} \phi(J, Q^{\hat{\pi}^{*(o)}}, \omega^{\hat{\pi}_l^{(o)},\nu}, \hat{\pi}_l^{(o)}).$$

Then it suffices to show the term in the first bracket converges to 0 faster than $\sqrt{nT}$. Notice that

$$\left\{ \phi(J, \widehat{Q}^{\hat{\pi}^{*(o)}}, \widehat{\omega}^{\hat{\pi}_l^{(o)},\nu}, \hat{\pi}_l^{(o)}) - \phi(J, Q^{\hat{\pi}^{*(o)}}, \omega^{\hat{\pi}_l^{(o)},\nu}, \hat{\pi}_l^{(o)}) \right.$$

$$\left. + (1-\gamma)\mathbb{E}_{S_0\sim\nu}[\sum_{a\in\mathcal{A}} \hat{\pi}_l^{(o)}(a|S_0)Q^{\hat{\pi}_l^{(o)}}(S_0, a)] - (1-\gamma)\mathbb{E}_{S_0\sim\nu}[\sum_{a\in\mathcal{A}} \hat{\pi}_l^{(o)}(a|S_0)Q^{\hat{\pi}_l^{(o)}}(S_0, a)] \right\}$$

$$= E_1 + E_2 + E_3,$$

where

$$E_1 = \frac{O}{nT} \sum_{(i,t)\in J_{o+1}} (\widehat{\omega}^{\hat{\pi}_l^{(o)},\nu}(S_{i,t},A_{i,t}) - \omega^{\hat{\pi}_l^{(o)},\nu}(S_{i,t},A_{i,t}))(R_{i,t} - Q^{\hat{\pi}_l^{(o)}}(S_{i,t},A_{i,t})$$

$$+\gamma \sum_{a\in\mathcal{A}} \hat{\pi}_l^{(o)}(a|S_{i,t+1})Q^{\hat{\pi}_l^{(o)}}(S_{i,t+1},a)),$$

$$E_2 = \frac{O}{nT} \sum_{(i,t)\in J_{o+1}} \omega^{\hat{\pi}_l^{(o)},\nu}(S_{i,t},A_{i,t})(\widehat{Q}^{\hat{\pi}_l^{(o)}}(S_{i,t},A_{i,t}) - Q^{\hat{\pi}_l^{(o)}}(S_{i,t},A_{i,t})$$

$$+\gamma \sum_{a\in\mathcal{A}} \hat{\pi}_l^{(o)}(a|S_{i,t+1})(\widehat{Q}^{\hat{\pi}_l^{(o)}}(S_{i,t+1},a) - Q^{\hat{\pi}_l^{(o)}}(S_{i,t+1},a))),$$

and

$$E_3 = \frac{O}{nT} \sum_{(i,t)\in J_{o+1}} (\widehat{\omega}^{\hat{\pi}_l^{(o)},\nu}(S_{i,t},A_{i,t}) - \omega^{\hat{\pi}_l^{(o)},\nu}(S_{i,t},A_{i,t}))(\widehat{Q}^{\hat{\pi}_l^{(o)}}(S_{i,t},A_{i,t}) - Q^{\hat{\pi}_l^{(o)}}(S_{i,t},A_{i,t})$$

$$+\gamma \sum_{a\in\mathcal{A}} \hat{\pi}_l^{(o)}(a|S_{i,t+1})(\widehat{Q}^{\hat{\pi}_l^{(o)}}(S_{i,t+1},a) - Q^{\hat{\pi}_l^{(o)}}(S_{i,t+1},a))).$$

Next, we bound each of the above three terms. For term $E_1$, it can be seen that

$$\mathbb{E}[E_1|\bar{J}_o] = 0.$$

In addition, by Assumptions 3 and 4, we can show

$$\mathrm{Var}[E_1] = \mathbb{E}[\mathrm{Var}(E_1|\bar{J}_o)] \lesssim \frac{O}{nT}(nT/O)^{-2\kappa_2},$$

where the inequality is based on that each item in $E_3$ is uncorrelated with others. Then by Markov's inequality, we can show

$$|E_1| = O_p((\frac{O}{nT})^{-1/2-\kappa_2}).$$

Similarly, we can show

$$|E_2| = O_p((\frac{O}{nT})^{-1/2-\kappa_1}).$$

For term $(E_3)$, by Cauchy Schwarz inequality and similar arguments as before, we can show

$$|E_3| = O_p((\frac{O}{nT})^{-(\kappa_2+\kappa_1)}).$$

Therefore, as long as $(\kappa_2 + \kappa_1) > 1/2$, we have $E_1 + E_2 + E_3 = o(\sqrt{O/nT})$, which concludes our proof.

**Proof of Theorem 1** We aim to show that

$$\frac{\sqrt{nT(O-1)/O}\left(\hat{\mathcal{V}}(\hat{\pi}_l) - \mathcal{V}(\hat{\pi}_l)\right)}{\hat{\sigma}(l)} \implies \mathcal{N}(0,1).$$

It can be seen that

$$\frac{\sqrt{nT(O-1)/O}\left(\hat{\mathcal{V}}(\hat{\pi}_l) - \mathcal{V}(\hat{\pi}_l)\right)}{\hat{\sigma}(l)} = \sqrt{\frac{nT}{O(O-1)}}\left(\sum_{o=1}^{O-1} \frac{\hat{\mathcal{V}}_{\mathcal{D}_{o+1}}(\hat{\pi}_l^{(o)}) - \mathcal{V}(\hat{\pi}_l)}{\hat{\sigma}_{o+1}(\hat{\pi}_l^{(o)})}\right)$$

$$= \sqrt{\frac{nT}{O(O-1)}}\left(\sum_{o=1}^{O-1} \frac{\hat{\mathcal{V}}_{\mathcal{D}_{o+1}}(\hat{\pi}_l^{(o)}) - \mathcal{V}(\hat{\pi}_l^{(o)})}{\hat{\sigma}_{o+1}(\hat{\pi}_l^{(o)})}\right)$$

$$+ \sqrt{\frac{nT}{O(O-1)}}\left(\sum_{o=1}^{O-1} \frac{\mathcal{V}(\hat{\pi}_l^{(o)}) - \mathcal{V}(\hat{\pi}_l)}{\hat{\sigma}_{o+1}(\hat{\pi}_l^{(o)})}\right).$$

Define

$$\phi(J, Q^\pi, w^\pi, \pi) = \frac{1}{|J|} \sum_{(i,t) \in J} w^{\pi,\nu}(S_{i,t}, A_t) \left( R_{i,t} + \gamma \sum_{a' \in \mathcal{A}} \pi(a'|S_{i,t+1}) Q^\pi(S_{9,t+1}, a') - Q^\pi(S_{i,t}, A_{i,t}) \right),$$

where $|J|$ is the cardinality of the index set $J$, i.e., $|J| = \frac{nT}{O}$. Then by Lemma 1, we show that

$$\sqrt{\frac{nT}{O}} \frac{\hat{\mathcal{V}}_{\mathcal{D}_{o+1}}(\hat{\pi}_l^{(o)}) - \mathcal{V}(\hat{\pi}_l^{(o)})}{\hat{\sigma}_{o+1}(\hat{\pi}_l^{(o)})} = \sqrt{\frac{nT}{O}} \frac{\phi(J_{o+1}, Q^{\hat{\pi}_l^{(o)}}, w^{\hat{\pi}_l^{(o)}}, \hat{\pi}_l^{(o)})}{\hat{\sigma}_{o+1}(\hat{\pi}_l^{(o)})} + o_p(1). \tag{16}$$

If we can show that

$$\max_{1 \leq o \leq (O-1)} \left| \frac{\hat{\sigma}_{o+1}(\hat{\pi}_l^{(o)})}{\sigma_{o+1}(\hat{\pi}_l^{(o)})} - 1 \right| = o_p(1),$$

which will be shown later, then by Slutsky theorem, we can show that

$$\sqrt{\frac{nT}{O(O-1)}} \left( \sum_{o=1}^{O-1} \frac{\hat{\mathcal{V}}_{\mathcal{D}_{o+1}}(\hat{\pi}_l^{(o)}) - \mathcal{V}(\hat{\pi}_l^{(o)})}{\hat{\sigma}_{o+1}(\hat{\pi}_l^{(o)})} \right)$$

$$= \underbrace{\sqrt{\frac{nT}{O(O-1)}} \left( \sum_{o=1}^{O-1} \frac{\phi(J_{o+1}, Q^{\hat{\pi}_l^{(o)}}, w^{\hat{\pi}_l^{(o)}}, \hat{\pi}_l^{(o)})}{\sigma_{o+1}(\hat{\pi}_l^{(o)})} \right)}_{(I)} + o_p(1).$$

For $(I)$, we can see that

$$(I) = \sqrt{\frac{O}{nT(O-1)}} \left( \sum_{o=1}^{O-1} \sum_{(i,t) \in J_{o+1}} w^{\hat{\pi}_l^{(o)},\nu}(S_{i,t}, A_{i,t})(R_{i,t} \tag{17}\right.$$

$$\left. + \gamma \sum_{a' \in \mathcal{A}} \hat{\pi}_l^{(o)}(a'|S_{i,t+1}) Q^{\hat{\pi}_l^{(o)}}(S_{i,t+1}, a') - Q^{\hat{\pi}_l^{(o)}}(S_{i,t}, A_{i,t})) / \sigma_{o+1}(\hat{\pi}_l^{(o)})). \tag{18}$$

By the sequential structure of our proposed algorithm, $(I)$ forms a mean zero martingale. Then we use Corollary 2.8 of (McLeish, 1974) to show its asymptotic distribution. First of all, by the uniformly bounded assumption on Q-function, ratio function and the variance, we can show that

$$\sqrt{\frac{O}{nT(O-1)}} \max_{1 \leq o \leq (O-1)} \max_{(i,t) \in J_0} \left| w^{\hat{\pi}_l^{(o)},\nu}(S_{i,t}, A_{i,t})(R_{i,t} + \gamma \sum_{a' \in \mathcal{A}} \hat{\pi}_l^{(o)}(a'|S_{i,t+1}) Q^{\hat{\pi}_l^{(o)}}(S_{i,t+1}, a') - \right.$$

$$\left. Q^{\hat{\pi}_l^{(o)}}(S_{i,t}, A_{i,t})) / \sigma_{o+1}(\hat{\pi}_l^{(o)}) \right| = o_p(1).$$

Next, we aim to show that

$$\frac{O}{nT(O-1)} \left| \left( \sum_{o=1}^{O-1} \sum_{(i,t) \in J_{o+1}} \{ w^{\hat{\pi}_l^{(o)},\nu}(S_{i,t}, A_{i,t})(R_{i,t} \right.\right. \tag{19}$$

$$\left.\left. + \gamma \sum_{a' \in \mathcal{A}} \hat{\pi}_l^{(o)}(a'|S_{i,t+1}) Q^{\hat{\pi}_l^{(o)}}(S_{i,t+1}, a') - Q^{\hat{\pi}_l^{(o)}}(S_{i,t}, A_{i,t}))\}^2 / \sigma_{o+1}^2(\hat{\pi}_l^{(o)}) \right) - 1 \right| = o_p(1).$$

Notice that the left hand side of the above is bounded above by

$$\frac{O}{nT} \max_{1 \leq o \leq (O-1)} \left| \left( \sum_{(i,t) \in J_{o+1}} \{ w^{\hat{\pi}_l^{(o)},\nu}(S_{i,t}, A_{i,t})(R_{i,t} \right.\right. \tag{20}$$

$$\left.\left. + \gamma \sum_{a' \in \mathcal{A}} \hat{\pi}_l^{(o)}(a'|S_{i,t+1}) Q^{\hat{\pi}_l^{(o)}}(S_{i,t+1}, a') - Q^{\hat{\pi}_l^{(o)}}(S_{i,t}, A_{i,t}))\}^2 / \sigma_{o+1}^2(\hat{\pi}_l^{(o)}) \right) - 1 \right|. \tag{21}$$

Because, for each $1 \le o \le (O-1)$,

$$\frac{O}{nT} \left\{ \left( \sum_{(i,t)\in J_{o+1}} \{ w^{\hat{\pi}_l^{(o)},\nu}(S_{i,t},A_{i,t})(R_{i,t} \right. \right. \tag{22}$$

$$+\gamma \sum_{a'\in\mathcal{A}} \hat{\pi}_l^{(o)}(a'|S_{i,t+1}) Q^{\hat{\pi}_l^{(o)}}(S_{i,t+1},a') - Q^{\hat{\pi}_l^{(o)}}(S_{i,t},A_{i,t}))\}^2 - \mathbb{E}[\{w^{\hat{\pi}_l^{(o)},\nu}(S,A)(R \tag{23}$$

$$\left. \left. +\gamma \sum_{a'\in\mathcal{A}} \hat{\pi}_l^{(o)}(a'|S') Q^{\hat{\pi}_l^{(o)}}(S',a') - Q^{\hat{\pi}_l^{(o)}}(S,A))\}]/\sigma_{o+1}^2(\hat{\pi}_l^{(o)})) \right\}, \tag{24}$$

forms a mean zero martingale, we apply Freedman's inequality in (Freedman, 1975) with Assumptions 3-5 to show it is bounded by $O_p(\sqrt{\frac{O}{nT}})$. Applying union bound shows (19) is $o_p(1)$ and furthermore consistency of $\hat{\sigma}(\hat{\pi}_l)$ in (16) holds. Then we apply the martingale central limit theorem to show

$$\sqrt{\frac{nT}{O(O-1)}} \left( \sum_{o=1}^{O-1} \frac{\phi(J_{o+1}, Q^{\hat{\pi}_l^{(o)}}, w^{\hat{\pi}_l^{(o)}}, \hat{\pi}_l^{(o)})}{\sigma_{o+1}(\hat{\pi}_l^{(o)})} \right) \Longrightarrow \mathcal{N}(0,1).$$

The remaining is to show

$$\sqrt{\frac{nT}{O(O-1)}} \left( \sum_{o=1}^{O-1} \frac{\mathcal{V}(\hat{\pi}_l^{(o)}) - \mathcal{V}(\hat{\pi}_l)}{\hat{\sigma}_{o+1}(\hat{\pi}_l^{(o)})} \right)$$

is asymptotically negligible. Consider

$$\mathbb{E}\left| \mathcal{V}(\hat{\pi}_l^{(o)}) - \mathcal{V}(\hat{\pi}_l) \right| \tag{25}$$

$$\le \mathbb{E}\left| \mathcal{V}(\hat{\pi}_l^{(o)}) - \mathcal{V}(\pi_l^*) \right| + \mathbb{E}\left| \mathcal{V}(\hat{\pi}_l) - \mathcal{V}(\pi_l^*) \right| \tag{26}$$

$$\le \mathbb{E}\left| \mathcal{V}(\hat{\pi}_l^{(o)}) - \mathcal{V}(\pi_l^*) \right| + \mathbb{E}\left| \mathcal{V}(\hat{\pi}_l) - \mathcal{V}(\pi_l^*) \right| \tag{27}$$

$$\le (nTo)^{-\kappa} O^\kappa + (nT)^{-\kappa}, \tag{28}$$

where we use Assumption 2 for the last inequality. Summarizing together, we can show that

$$\sqrt{\frac{nT}{O(O-1)}} \mathbb{E}\left| \sum_{o=1}^{O-1} \mathcal{V}(\hat{\pi}_l^{(o)}) - \mathcal{V}(\hat{\pi}_l) \right|$$

$$\le \sqrt{\frac{nT}{O(O-1)}} \sum_{o=1}^{O-1} (nTo)^{-\kappa} O^\kappa + \sqrt{\frac{nT(O-1)}{O}} (nT)^{-\kappa}$$

$$\le \sqrt{\frac{nTO^2}{O(O-1)}} \sum_{o=1}^{O-1} (nT)^{-\kappa} + \sqrt{\frac{nT(O-1)}{O}} (nT)^{-\kappa}$$

$$= o(1),$$

where we obtain the second inequality by that $\sum_{o=1}^{O-1} o^{-\kappa} \le 1 + \int_1^O o^{-\kappa} do \lesssim O^{1-\kappa}$. In the last inequality, we use $\kappa > 1$ in Assumption 2. Then Markov inequality gives that

$$\sqrt{\frac{nT}{O(O-1)}} \left( \sum_{o=1}^{O-1} \mathcal{V}(\hat{\pi}_l^{(o)}) - \mathcal{V}(\hat{\pi}_l) \right) = o_p(1).$$

Moreover, by Assumption 5 that $\inf_{1 \le o \le O-1} \hat{\sigma}_{o+1}(\hat{\pi}_l^{(o)}) \ge c$ for some constant $c > 0$, we can further show that

$$\sqrt{\frac{nT}{O(O-1)}} \left( \sum_{o=1}^{O-1} \frac{\mathcal{V}(\hat{\pi}_l^{(o)}) - \mathcal{V}(\hat{\pi}_l)}{\hat{\sigma}_{o+1}(\hat{\pi}_l^{(o)})} \right) = o_p(1),$$

which completes our proof.

**Proof of Corollary 1** Denote the sets $E_l = \{|\mathcal{V}(\hat{\pi}_l) - \hat{\mathcal{V}}(\hat{\pi}_l)| \leq \hat{u}(l)\}$, $l = 1, \ldots, L$, where $\hat{u}(l) = z_{\alpha/2}\sqrt{nT(O-1)/O}\hat{\sigma}(l)$. Note that $\liminf_{nT\to\infty} \Pr(\cap_{j=1}^L E_j) \geq 1 - L\alpha$ and

$$\Pr(\mathcal{V}(\hat{\pi}_{\hat{l}}) \geq \max_{1\leq l\leq L} \mathcal{V}(\hat{\pi}_l) - 2\hat{u}(l))$$

$$= \Pr(\mathcal{V}(\hat{\pi}_{\hat{l}}) - \hat{\mathcal{V}}(\hat{\pi}_{\hat{l}}) + \hat{\mathcal{V}}(\hat{\pi}_{\hat{l}}) \geq \max_{1\leq l\leq L} \mathcal{V}(\hat{\pi}_l) - \hat{\mathcal{V}}(\hat{\pi}_l) - 2\hat{u}(l) + \hat{\mathcal{V}}(\hat{\pi}_l))$$

$$\geq \Pr(\mathcal{V}(\hat{\pi}_{\hat{l}}) - \hat{\mathcal{V}}(\hat{\pi}_{\hat{l}}) + \hat{\mathcal{V}}(\hat{\pi}_{\hat{l}}) \geq \max_{1\leq l\leq L} \mathcal{V}(\hat{\pi}_l) - \hat{\mathcal{V}}(\hat{\pi}_l) - 2\hat{u}(l) + \hat{\mathcal{V}}(\hat{\pi}_l)| \cap_{j=1}^L E_j) \Pr(\cap_{j=1}^L E_j)$$

$$\geq \Pr(\hat{\mathcal{V}}(\hat{\pi}_{\hat{l}}) - \hat{u}(\hat{l}) \geq \max_{1\leq l\leq L} \hat{\mathcal{V}}(\hat{\pi}_l) - \hat{u}(l)| \cap_{j=1}^L E_j) \Pr(\cap_{j=1}^L E_j)$$

$$= \Pr(\cap_{j=1}^L E_j),$$

where the last inequality holds because given the event $\cap_{j=1}^L E_j$, one has $-\hat{u}(\hat{l}) \leq \mathcal{V}(\hat{\pi}_{\hat{l}}) - \hat{\mathcal{V}}(\hat{\pi}_{\hat{l}})$ and $\mathcal{V}(\hat{\pi}_l) - \hat{\mathcal{V}}(\hat{\pi}_l) \leq \hat{u}(l)$ for any $l$. This completes the proof by taking $\liminf$ on both sides.

**Proof of Theorem 2** To show the results in Theorem 2, it can be seen that

$$\left|\frac{\sqrt{nT(O-1)/O}\left(\hat{\mathcal{V}}(\hat{\pi}_l) - \mathcal{V}(\pi^*)\right)}{\hat{\sigma}(l)}\right| \leq \left|\sqrt{\frac{nT}{O(O-1)}}\left(\sum_{o=1}^{O-1} \frac{\hat{\mathcal{V}}_{\mathcal{D}_{o+1}}(\hat{\pi}_l^{(o)}) - \mathcal{V}(\hat{\pi}_l^{(o)})}{\hat{\sigma}_{o+1}(\hat{\pi}_l^{(o)})}\right)\right|$$

$$+ \sqrt{\frac{nT}{O(O-1)}}\left(\sum_{o=1}^{O-1} \frac{\mathcal{V}(\hat{\pi}_l^{(o)}) - \mathcal{V}(\pi^*)}{\hat{\sigma}_{o+1}(\hat{\pi}_l^{(o)})}\right)$$

$$\leq \underbrace{\left|\sqrt{\frac{nT}{O(O-1)}}\left(\sum_{o=1}^{O-1} \frac{\hat{\mathcal{V}}_{\mathcal{D}_{o+1}}(\hat{\pi}_l^{(o)}) - \mathcal{V}(\hat{\pi}_l^{(o)})}{\hat{\sigma}_{o+1}(\hat{\pi}_l^{(o)})}\right)\right|}_{(I)}$$

$$+ B(l)\sqrt{\frac{nT}{O(O-1)}}\left(\sum_{o=1}^{O-1} \frac{1}{\hat{\sigma}_{o+1}(\hat{\pi}_l^{(o)})}\right).$$

Then by results in the proof of Theorem 1, we can show that

$$\lim_{nT\to\infty} \Pr((I) > z_{\alpha/2}) = \alpha. \tag{29}$$

This implies that

$$\liminf_{nT\to\infty} \Pr\left(|\mathcal{V}(\pi^*) - \hat{\mathcal{V}}(\hat{\pi}_l)| \leq z_{\alpha/2}\sqrt{O/nT(O-1)}\hat{\sigma}(l) + B(l)\right) \tag{30}$$

$$\geq \lim_{nT\to\infty} \Pr((I) \leq z_{\alpha/2}) = 1 - \alpha, \tag{31}$$

which concludes our proof.

**Proof of Corollary 2**: We mainly show the proof of the second claim in the corollary, based on which the first claim can be readily seen. Define an event $E$ such that $1 \leq l \leq L$, $|\mathcal{V}(\hat{\pi}_l) - \hat{\mathcal{V}}(\hat{\pi}_l)| \leq c(\delta)\log(L)\hat{\sigma}(i)/\sqrt{NT}$ and $|\mathcal{V}(\pi^*) - \hat{\mathcal{V}}(\hat{\pi}_l)| \leq z_{\alpha/(2L)}\sqrt{O/nT(O-1)}\hat{\sigma}(l) + B(l)$. Based on the assumption given in Corollary 2 and Theorem 2, we have $\liminf_{nT\to\infty} P(E) \geq 1-\delta-\alpha$. In the following, we suppose event $E$ holds.

Inspired by the proofs of Corollary 1 in (Mathé, 2006) and Theorem 3 of (Su et al., 2020), we define $\tilde{l} = \max\{l : B(l) \leq u_1(l) + u_2(l)\}$, where $u_1(l) = z_{\alpha/(2L)}\sqrt{O/nT(O-1)}\hat{\sigma}(l)$. Let $u_2(l) = c(\delta)\log(L)\hat{\sigma}(i)/\sqrt{NT}$. By Assumption 7, for $l \leq \tilde{l}$,

$$B(l) \leq B(\tilde{l}) \leq u_1(\tilde{l}) \leq u_1(l),$$

which further implies that for any $l \leq \tilde{l}$,

$$|\hat{\mathcal{V}}(\hat{\pi}_l) - \mathcal{V}(\pi^*)| \leq B(l) + u_1(l) \leq 2u_1(l).$$

Then $\mathcal{V}(\pi^*) \in I(l)$ based on the construction of $I(l)$ for all $l \leq \tilde{l}$. In addition, we have for $l \leq \tilde{l}$

$$|\mathcal{V}(\hat{\pi}_l) - \mathcal{V}(\pi^*)| \leq 2u_1(l) + u_2(l), \tag{32}$$

by triangle inequality and event $E$. Since $I(l)$ share at least one common element for $1 \leq l \leq \tilde{l}$, we have $\hat{i} \geq \tilde{l}$. Moreover, there must exist an element $x$ such that $x \in I(\tilde{l}) \cap I(\hat{i})$, where $|\hat{\mathcal{V}}(\hat{\pi}_{\tilde{l}}) - x| \leq u_1(\tilde{l})$ and $|\hat{\mathcal{V}}(\hat{\pi}_{\hat{i}}) - x| \leq u_1(\hat{i})$. This indicates that

$$|\hat{\mathcal{V}}(\hat{\pi}_{\hat{i}} - \mathcal{V}(\pi^*)| \leq |\hat{\mathcal{V}}(\hat{\pi}_{\hat{i}}) - x| + |\hat{\mathcal{V}}(\hat{\pi}_{\tilde{l}}) - x| + |\hat{\mathcal{V}}(\hat{\pi}_{\tilde{l}}) - \mathcal{V}(\pi^*)| \tag{33}$$
$$\leq u_1(\hat{i}) + 2u_1(\tilde{l}) \leq 3u_1(\tilde{l}), \tag{34}$$

by again triangle inequality and Assumption 7, and

$$|\mathcal{V}(\hat{\pi}_{\hat{i}}) - \mathcal{V}(\pi^*)| \leq u_2(\hat{i}) + 3u_1(\tilde{l}) \leq u_2(\tilde{l}) + 3u_1(\tilde{l}), \tag{35}$$

by event $E$ and Assumption 7. Define $l^* = \min\{l : B(l) + u_1(l) + u_2(l)\}$. Then following the similar proof of (Su et al., 2020), we consider two cases:

**Case 1:** If $l^* \leq \tilde{l}$, then we have

$$u_2(\tilde{l}) + B(\tilde{l}) + u_1(\tilde{l}) \leq 2u_1(l^*) + u_2(l^*) \leq 2u_1(l^*) + 2B(l^*) + u_2(l^*),$$

where we use Assumption 7.

**Case 2:** If $l^* > \tilde{l}$, then we have

$$\zeta(u_2(\tilde{l}) + u_1(\tilde{l})) \leq (u_2(\tilde{l}+1) + u_1(\tilde{l}+1)) \leq B(\tilde{l}+1) \leq B(l^*),$$

where we use Assumption 7. This implies that

$$u_2(\tilde{l}) + u_1(\tilde{l}) + B(\tilde{l}) \leq (1 + 1/\zeta)B(l^*).$$

Combining two cases, we can show that

$$u_2(\tilde{l}) + u_1(\tilde{l}) + B(\tilde{l}) \leq (1 + 1/\zeta)(B(l^*) + u_1(l^*) + u_2(l^*)),$$

as $\zeta < 1$. Together with (33), we have

$$|\mathcal{V}(\hat{\pi}_{\hat{i}}) - \mathcal{V}(\pi^*)| \leq u_2(\hat{i}) + 3u_1(\tilde{l}) \leq 3(1 + 1/\zeta)(B(l^*) + u_1(l^*) + u_2(l^*)), \tag{36}$$

which concludes our proof.

## C  MORE DETAILS ON DQN ENVIRONMENTS

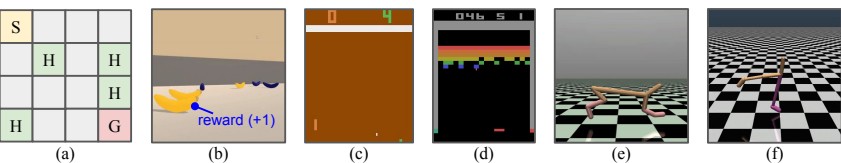

(a)  (b)  (c)  (d)  (e)  (f)

**Figure 5:** DQN environments in our studies: (a) $\mathbf{E}_1$: *FrozenLake-v0*; (b) $\mathbf{E}_2$: *Banana Collectors* (3D geometrical navigation task); (c) $\mathbf{E}_3$: *Pong-v0*; (d) $\mathbf{E}_4$: *Breakout-v0*; (e) $\mathbf{E}_5$: *Halfcheetah-v1*; (f) $\mathbf{E}_6$: *Walker2d-v1*.

We introduce our deployed DQN environments in this section, which included four environments with discrete action ($\mathbf{E}_1$ to $\mathbf{E}_4$) and two environments ($\mathbf{E}_5$ to $\mathbf{E}_6$) with continuous action. These environments cover wide applications, including tabular learning ($\mathbf{E}_1$), navigation to a target object in a geometrical space ($\mathbf{E}_2$), digital gaming ($\mathbf{E}_3$ to $\mathbf{E}_4$), and continuous control ($\mathbf{E}_5$ to $\mathbf{E}_6$).

$\mathbf{E}_1$**: Frozen Lake:** The Frozen Lake is a maze environment that manipulates an agent to walk from a starting point (S) to a goal point without failing into the hole (H). We use *FrozenLake-v0* from OpenAI Gym (Brockman et al., 2016) as shown in Fig 5(a). We provide top-5 regret and precision results shown in Figure 6 and 7.

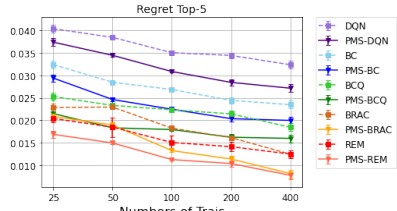 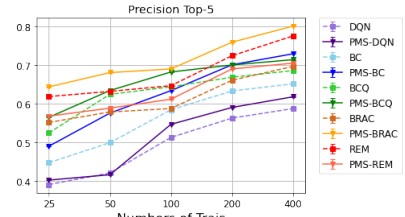

**Figure 6:** Policy selection using top-k ranking regret score in $\mathbf{E}_1$ (Frozen Lake).

**Figure 7:** Policy selection using top-k ranking precision in $\mathbf{E}_1$ (Frozen Lake).

$\mathbf{E}_2$: **Banana Collector:** The Banana collector is one popular 3D-graphical navigation environment that compresses discrete actions and states as an open source DQN benchmark from Unity [1] ML-Agents v0.3.(Juliani et al., 2018). The DRL agent controls an automatic vehicle with 37 dimensions of state observations including velocity and a ray-based perceptional information from objects around the agent. The targeted reward is $12.0$ points by accessing correct yellow bananas $(+1)$ and avoiding purple bananas $(-1)$ in first-person point of view as shown in Fig 5(b). We provide the related top-5 regret and precision results shown in Figure 8 and 9.

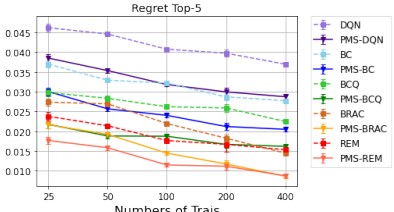 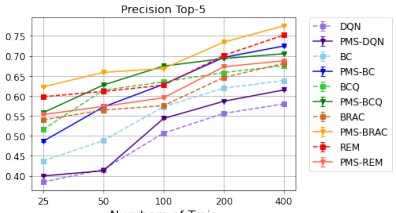

**Figure 8:** Policy selection using top-k ranking regret score in $\mathbf{E}_2$ (Banana Collector).

**Figure 9:** Policy selection using top-k ranking precision in $\mathbf{E}_2$ (Banana Collector).

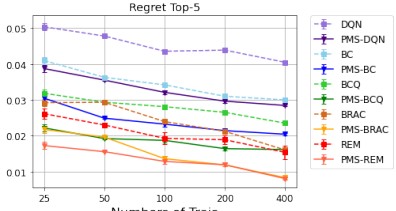 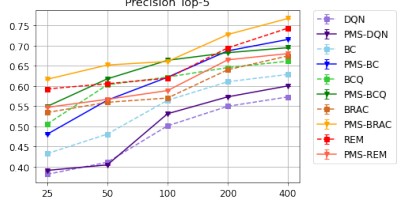

**Figure 10:** Policy selection using top-k ranking regret score in $\mathbf{E}_3$ (Pong).

**Figure 11:** Policy selection using top-k ranking precision in $\mathbf{E}_3$ (Pong).

$\mathbf{E}_3$: **Pong:** Pong is one Atari game environment from OpenAI Gym (Brockman et al., 2016) as shown in Fig 5(c). We provide its top-5 regret and precision results shown in Figure 10 and 11.

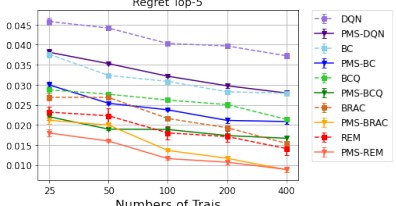 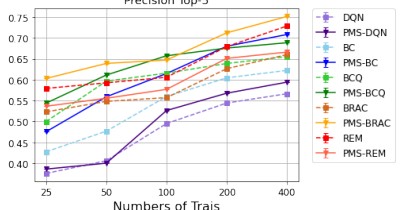

**Figure 12:** Policy selection using top-k ranking regret score in $\mathbf{E}_4$ (Breakout).

**Figure 13:** Policy selection using top-k ranking precision in $\mathbf{E}_4$ (HalfCheetah-v1).

$\mathbf{E}_4$: **Breakout:** Breakout is one Atari game environment from OpenAI Gym (Brockman et al., 2016) as shown in Fig 12(d). We provide the related top-5 regret and precision results shown in Figure 12 and 13.

---

[1] https://www.youtube.com/watch?v=heVMs3t9qSk

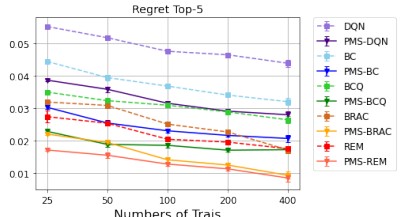

**Figure 14:** Policy selection using top-k ranking regret score in $\mathbf{E}_5$ (HalfCheetah-v1).

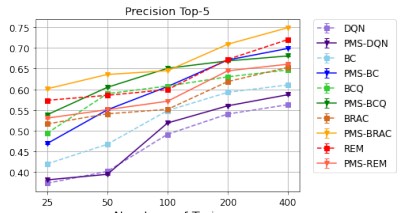

**Figure 15:** Policy selection using top-k ranking precision in $\mathbf{E}_5$ (HalfCheetah-v1).

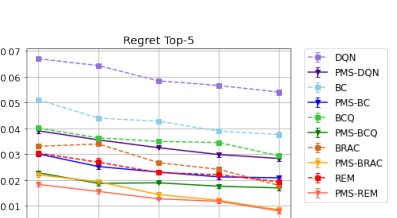

**Figure 16:** Policy selection using top-k ranking regret score in $\mathbf{E}_6$ (Walker2d-v1).

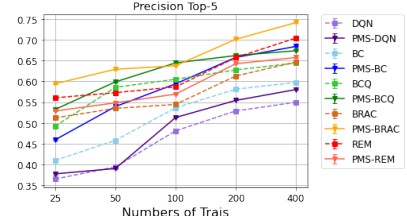

**Figure 17:** Policy selection using top-k ranking precision in $\mathbf{E}_6$ (Walker2d-v1).

$\mathbf{E}_5$: **HalfCheetah-v1:** Halfcheetah is a continuous action and state environment to control agent with monuments made by MuJoCo simulators as shown in Fig 5(e). We provide the related top-5 regret and precision results shown in Figure 14 and 15.

$\mathbf{E}_6$: **Walker2d-v1:** Walker2d-v1 is a continuous action and state environment to control agent with monuments made by MuJoCo simulators as shown in Fig 5(f). We provide the related top-5 regret and precision results shown in Figure 16 and 17.

### C.1 HYPER-PARAMETERS INFORMATION

We select a total of 70 DQN based models for each environment. We will open source the model and implementation for future studies. Table 1, Table 2, and Table 3 summarize their hyper-parameter and setups. In addition, Figure 18 and Figure 19 provide ablation studies on different scales of $\alpha$ and $O$ selection in PMS experiments for the deployed DRL navigation task ($\mathbf{E}_2$). From the experimental results, a more pessimistic $\alpha$ (e.g., 0.001) is associated with a slightly better attained top-5 regret. Meanwhile, the selection of $O$ does not produce much different performance on selected policies but slightly affects the range of the selected policies.

**Table 1:** Hyper-parameters information for for DQN models used in $\mathbf{E}_1$ to $\mathbf{E}_2$

| Hyper-parameters | Values |
| --- | --- |
| Hidden layers | $\{1, 2\}$ |
| Hidden units | $\{16, 32, 64, 128\}$ |
| Learning rate | $\{1 \times e^{-3}, 5 \times e^{-4}\}$ |
| DQN training iterations | $\{100, 500, 1k, 2k\}$ |
| Batch size | $\{64\}$ |

### BROADER IMPACT

There are also some limitations of the proposed PMS as one of the preliminary attempts on model selection for offline reinforcement learning. When the benchmarks environments (excluded Atari games) are based on simulated environments to collect the true policy (Barth-Maron et al., 2018; Siegel et al., 2019), more real-world-based environments could be customized and studied in future works. For example, one experimental setup needs to be carefully controlled in clinical settings (Tang & Wiens, 2021) or resilience-oriented (Yang et al., 2021) reinforcement learning.

**Table 2:** Hyper-parameters information for for DQN models used in $\mathbf{E}_3$ to $\mathbf{E}_4$

| Hyper-parameters | Values |
|---|---|
| Convolutional layers | $\{\,2,3\}$ |
| Convolutional units | $\{16,\ 32\}$ |
| Hidden layers | $\{\,2,3\}$ |
| Hidden units | $\{64,\ 256,\ 512\}$ |
| Learning rate | $\{1 \times \mathrm{e}^{-3},\ 5 \times \mathrm{e}^{-4}\}$ |
| DQN training iterations | $\{4M,\ 4.5M,\ 5M\}$ |
| Batch size | $\{64\}$ |

**Table 3:** Hyper-parameters information for double DQN (DDQN) models (Van Hasselt et al., 2016) with a prioritized replay (Schaul et al., 2015) used in $\mathbf{E}_5$ to $\mathbf{E}_6$.

| Hyper-parameters | Values |
|---|---|
| Hidden layers | $\{4,\ 5,\ 6\}$ |
| Hidden units | $\{64,\ 128,\ 256,\ 512\}$ |
| Learning rate | $\{1 \times \mathrm{e}^{-3},\ 5 \times \mathrm{e}^{-4}\}$ |
| DDQN training frames | $\{40M,\ 45M,\ 50M\}$ |
| Batch size | $\{256\}$ |
| Buffer size | $\{10^6\}$ |
| Updated target | $\{1000\}$ |

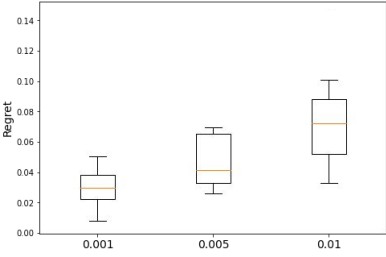

**Figure 18:** Different $\alpha$ for PMS selection.

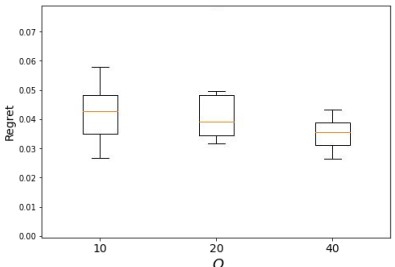

**Figure 19:** Different $O$ for PMS selection.

