# OpenReview forum: "Pessimistic Model Selection for Offline Deep Reinforcement Learning"
_ICLR.cc/2022/Conference — ICLR 2022 Submitted_

### Official Review · Reviewer_MuG4 · 2021-10-19

**Correctness:** 4
**Technical Novelty And Significance:** 3
**Empirical Novelty And Significance:** 2
**Recommendation:** 5
**Confidence:** 3

**Main Review:**

**Strengths**
- the paper tackles the important problem of model selection in ORL
- the proposed method is based on religious guarantees, there are also two refined approaches for addressing the issue in an assumption
- the simulation study shows the benefit of using the proposed approach

**Weaknesses**
- The claim “our approach essentially does not involve additional hyperparameter tuning” seems somewhat overselling. We should specify “O” and “\alpha” to use PMS, and how they affect the model selection performance is unclear in the experiment

- Some related papers are missing such as [a] and [b]

[a] Mengjiao Yang, Bo Dai, Ofir Nachum, George Tucker, and Dale Schuurmans. Offline Policy Selection under Uncertainty. https://arxiv.org/abs/2012.06919

[b] Ilja Kuzborskij, Claire Vernade, András György, and Csaba Szepesvári. Confident Off-Policy Evaluation and Selection through Self-Normalized Importance Weighting. https://arxiv.org/abs/2006.10460

- there is room for improvement in the experimental section
    - regret is defined in an additive manner, and Figure 3 contains results across different environments. Is this valid? Using relative regret ($\frac{\mathcal{V}\left(\pi_{l^{*}}\right)-\mathcal{V}\left(\pi_{l}\right)}{\mathcal{V}\left(\pi_{l^{*}}\right)}$) seems more reasonable.
    - What each point in Figure 4 means seems unclear
    - I would love to see how the choice of “O” and “\alpha” affect the model selection of PMS
    - Figures 1 and 5 contain the result from only one of six environments. It is possible to use Appendix to share the results from the other environments that are not presented in the paper currently
    - Additional evaluation metrics such as top-k presion used in [b] would make the experiments more practically relevant

I would consider raising my score if I see substantial improvement in the experimental section. It is also essential to clarify the contributions of the paper compared to [a] and [b].


**Summary Of The Paper:**

The paper studies the model selection problem in offline reinforcement learning (ORL). The model selection of ORL is challenging due to insufficient observational data from offline collection. The paper proposes a novel model selection approach to automate ORL development process and to identify a well-performed model given offline data. The proposed pessimistic model selection (PMS) method leverages an uncertainty quantification on value functions and a pessimistic idea. Some experiments show the superior performance of the proposed model selection method.

**Summary Of The Review:**

The paper studies an important problem and provide nice solutions. However, some important related papers are missing and there are number of concerns in the experiment. Thus, I would recommend weak reject at this moment. I would like to see substantial effort, in particular the improvement in the experiments to change my score.

---

> ### Author Response · Authors · 2021-11-22
> **Reply to Reviewer MuG4**
>
>
> Thank you for your understanding of the novelty of our work and for providing related suggestions.
> We summarized the related questions below.
>
> ***
>
> **Q1:** Related Offline RL References
>
> **A1.** Thank you for the suggestion. We have incorporated these two important references in our revised version.
>
> ***
>
> **Q2:** Regret presentation
>
> **A2:** We apologize for this confusion and thank the reviewer for providing the suggestion. In Figure 3, we provide a reference responding to the performance studies in Figure 1 for Env_2 (navigation). We provide more per-environment studies in Figure 6 to Figure 17 with top-k relative regret and precision results for each environment.
>
> ***
>
> **Q3.** I would love to see how the choice of “O” and “$\alpha$” affect the model selection of PMS
>
> **A3.** Thank you for the suggestion and this interest. We provide additional results, including different choices of “O” and “$\alpha$” in Appendix Figure 18 and 19. In summary, a smaller $\alpha$ could attain slightly better performance in terms of top-5 regrets. Meanwhile, the selection of “O” does not affect the performance much. (within a relative 0.07 standard deviation)
>
> ***

---

### Official Review · Reviewer_Y9ri · 2021-10-24

**Correctness:** 3
**Technical Novelty And Significance:** 2
**Empirical Novelty And Significance:** 2
**Recommendation:** 3
**Confidence:** 3

**Main Review:**

High-level comments:
1. The theoretical part is weak, with many strong assumptions being made, but only providing asymptotic convergence results.
2. The exposition can be significantly improved. The proposed method is based on many existing techniques like doubly-robust estimators, for which little explanation is provided. No self-contained pseudo-code is provided. Many notations are used without definition.

Detailed comments:

1. Please avoid using the acronym ORL for offline reinforcement learning, as online reinforcement learning can also be abbreviated with the same acronym.

2. In related works, when discussing other works on model-selection in offline RL, please elaborate on their techniques and contrast with the current paper.

3. $d^{\pi,\nu}$ is not defined.

4. Could the authors elaborate on how equation (4) comes about? The RHS doesn't depend on the MDP transition at all, whereas the LHS clearly does, so how could they be equal?

5. Section 4.1, I think you mean "an estimate of $Q^{\hat\pi_\ell}$ as $\hat Q_\ell$", since there is no assumption that any of the $\pi_\ell$ is close to the optimal policy.

6. Sec 4.2 in the description of the proposed algorithm, above equation (6), what do you mean by "applying FQI on $D_0$ to compute $\hat Q_\ell$"? To my knowledge, FQI only take as input a value function class $\mathcal{F}$ but doesn't take any policy as additional input. If you mean to evaluate a particular policy $\pi_\ell$, the procedure is called Fitted-Q Evaluation (FQE). Also, it is confusing what "candidate model" means. Does it mean candidate policies returned by L different algorithms? The phrase "model" is very often confusing in RL, as many constructs are referred to as models, e.g. MDP transition in model-based RL or model as in a trained neural network. Avoid it if you can and be specific.


7. $\hat Q^{\star (o)}_\ell$ in (8) is not defined.

8. What is the justification for constructing the confidence interval with finite samples based on an asymptotic convergence result? In finite sample regime, (11) can be arbitrarily far off.

9. In what way can we see that large $O$ increases accuracy? Solely based on equation (11), the effect of large O on the convergence rate is negligible.

10. $\alpha$ is defined multiple times, in section 4 as the confidence parameter and in section 5 assumption 2 as the convergence rate.

11. For section 6, could the authors elaborate on how their refined approach automatically adapts to unknown biases $B(\ell)$? Also if there is bias in assumption 2 for policy improvement, then there definitely should be biases in assumption 3 for policy evaluation. Then, the additional assumption in Corollary 2 should not hold.

12. No discussion about the computational complexity of the proposed algorithm. In particular, the min-max problem in doubly-robust estimation seems to scale poorly.

**Summary Of The Paper:**

This paper proposed a new algorithm for model-section in offline RL based on adding the pessimism principle on top of existing OPE algorithms. Under strong assumptions, the paper provides an asymptotic convergence result for the proposed algorithm. It also provides empirical experiments demonstrating that the proposed algorithm outperforms existing baselines.

**Summary Of The Review:**

The theoretical result is weak, and the exposition can be significantly improved.

---

> ### Author Response · Authors · 2021-11-22
> **Reply to Reviewer Y9ri (1/2)**
>
> Thank you for providing suggestions and theoretical discussions on our work. We summarized the related questions below.
>
> ***
>
> **Q1:** The theoretical part is weak, with many strong assumptions being made, but only providing asymptotic convergence results.
>
> **A1:** We are sorry to learn that the reviewer felt our theoretical part is weak. We would like to provide more explanations below to illustrate the significance of our work. Having a valid confidence interval is a challenging task; even asymptotics are not easy to obtain (e.g., temporal dependence, non-regularity caused by possibly non-unique optimal policy). Knowing an asymptotic result will bring some benefit for our model selection framework. Our algorithm based on the asymptotic convergence result is general for any outcome model and comes with high practical utility when the sample size is relatively large. Having an exact confidence interval might require some model specification of the value function, and using non-asymptotic bounds might require additional tuning steps (e.g., constants in many concentration inequalities), which is beyond the scope of this paper. Moreover, methodologically speaking, our algorithm is able to incorporate exact confidence intervals and non-asymptotic bounds. We have added some discussions related to this in Appendix A.
>
> ***
>
> **Q2:** The exposition can be significantly improved. The proposed method is based on many existing techniques like doubly-robust estimators, for which little explanation is provided. No self-contained pseudo-code is provided. Many notations are used without definition.
>
> **A2:** More explanations are added in the revised version, and the algorithm is revised to be self-contained. We also note that the proposed selection framework is generic in the sense that it does not only apply to doubly-robust estimators.
>
>
> ***
>
> **Q3:** Please avoid using the acronym ORL for offline reinforcement learning, as online reinforcement learning can also be abbreviated with the same acronym.
>
> **A3:** We have revised the paper by using "OffRL."
>
>
> ***
>
> **Q4:** In related works, when discussing other works on model selection in offline RL, please elaborate on their techniques and contrast with the current paper.
>
> **A4:** Following the reviewer’s suggestion, we have revised related works.
>
> ***
>
> **Q5:** Section 4.1 discussion
>
> **A5:** When using FQI, one wishes to use $\widehat Q_l$ to approximate $Q^\ast$. This is why we say $\widehat Q_l$ is an estimate of $Q^\ast$, while it could be biased, as the reviewer said.
>
> ***
>
> **Q6** Discussion related to FQI
>
> **A6:** Candidate models can be different functional classes for modeling the optimal Q-function. Once we choose a specific model to run FQI, we can obtain an estimate for $Q^\ast$, denoted by $\widehat Q_l$. Correspondingly, we can get an estimated optimal policy $\hat \pi^{(l)}$ by letting $\pi^{(l)}(a | s) \in \text{argmax}_{a \in A} \widehat Q_l(s, a)$. In this setting, it is equivalent to having L candidate policies. We do not run FQE in our procedure (We could further run FQE to improve the evaluation of $\hat \pi^{(l)}$). In addition, thanks for pointing out the confusion of using “models”. In order to make it more clear, we have added some explanation about what models refer to when we first use this terminology.
>
> ***

---

> > ### Author Response · Authors · 2021-11-22
> > **Reply to Reviewer Y9ri (2/2)**
> >
> > **Q7:** What is the justification for constructing the confidence interval with finite samples based on an asymptotic convergence result? In finite sample regime, (11) can be arbitrarily far off.
> >
> > **A7:** All theoretical justification in this paper is based on asymptotics. It might be possible to investigate finite sample regimes when one has an exact confidence interval instead of (12) or a non-asymptotic bound. However, having an exact confidence interval might require some strong model specification of the value function, and using non-asymptotic bounds might require additional tuning steps, which is beyond the scope of this paper. In addition, as can be seen from empirical evaluations, with a relatively large sample size, the proposed model selection approach performs well. Nevertheless, we agree that if the sample size is small and the outcome is highly skewed, our algorithm based on (12) can be off. However, the sensitivity result in Fig. 4 shows that our result still performs better than other methods in this regime.
> >
> > ***
> >
> > **Q8:** The effect of large $O$ on the convergence rate is negligible.
> >
> > **A8:** You are right; when $O$ is large, it is negligible. The accuracy is discussed in terms of finite-sample performance. We believe the performance may be discounted by $(O-1)/O$ and therefore, the increase of $O$ will increase the accuracy.
> >
> > ***
> >
> >
> > **Q9**: $\alpha$ definition issues.
> >
> > **A9:** Appreciate your suggestion and we have fixed this issue.
> >
> > ***
> >
> > **Q10** how their refined approach automatically adapts to unknown biases
> >
> > **A10:** Thanks for your insightful comments. For our first refined approach, after sorting models by the standard errors from high to low, we assume the biases of sorted models are from low to high due to the bias-variance tradeoff. In this case, in the beginning, standard errors dominate biases. Then we show under some mild conditions; the true value can be covered by the confidence interval $I(l)$ defined after Theorem 2 when $l$ is small. Among all these models, we should choose the one with the smallest standard error. This is why we choose the largest $j$ such that models up to $j$ have overlapping confidence intervals. For the second approach, instead of choosing the smallest standard error, we choose the one with the best worst performance among candidate models. If Assumption 2 does not hold, it is possible that Assumption 3 may not hold either. However, in the current analysis, such an assumption is the best we can make. We plan to relax this assumption in the future. One possible way is to additionally run FQE on each obtained policy.
> >
> > ***
> >
> > **Q11** No discussion about the computational complexity of the proposed algorithm. In particular, the min-max problem in doubly-robust estimation seems to scale poorly.
> >
> > **A11:** You are right; min-max optimization may not be scaled well. However, such a method for ratio function estimation is indeed state-of-the-art. The computational complexity of solving this min-max problem is the number of outer iterations times the number of inner iterations times the cost of calculating the stochastic gradient.

---

### Official Review · Reviewer_Sxiw · 2021-10-30

**Correctness:** 3
**Technical Novelty And Significance:** 2
**Empirical Novelty And Significance:** 2
**Recommendation:** 3
**Confidence:** 4

**Main Review:**

This submission presented a pessimistic model selection approach (PMS) for offline deep reinforcement learning (ORL) and tested the approach with performance comparison on six simulated environments. Under several specified conditions, the authors provided asymptotic analysis to show that PMS can lead to the best model with respect the worst performance of the derived policy.

**Strengths:**

1. The authors proposed the PMS to select ORL model based on the constructed U(l) confidence interval estimate (page 6) by splitting offline training data and sequential estimation, which has the potential to address overfitting issues. Both theoretical analysis and empirical performance have been conducted.

2. Two refined approaches were presented considering potential bias, even though empirically not showing advantages in simulated environments.

**Weaknesses:**

1. The authors may want to better explain some steps of the proofs in the appendix, for example, the last inequality for the proof of corollary 1 on page 17.

2. Regarding empirical evaluation of PMS, it appears that the theoretical analyses were mostly based on several references, including Su et al (2020). In addition to comparing to OPE estimates in Tang & Wiens (2021), can the authors compare with other competing methods? Also, why the authors did not show the corresponding regrets for other competing methods in Figure 3 to better show the advantages of PMS? In Figure 3, what is $\pi^*$? If it is the optimal policy, shouldn't the corresponding regret be zero? Also by simply looking at the tables in the appendix, it is not really clear how the initial models were selected in different experiments. More comprehensive performance evaluation should be provided.

3. The submission may have to be checked carefully. For example, it is clear that the indices do not match in the last equation on page 7 for the constructed interval: should the index to be i instead of l in the righthand side?  In the beginning paragraph of Appendix A, the inequality directions are not consistent. The starting indices for t (from page 3) and o (from page 5) have to be checked to make them correct.

4. The presentation needs to be improved. There are many language problems. For example, several acronyms were never defined, including OPE, which I assume that it stands for Offline Policy Evaluation. There are many grammar errors, for example, "such A guarantee" on page 2; "such A procedure" on page 8; "...in continuous caseS" on page 3. There are also incomplete sentences, for example, "...where the refinement algorithms (PMS R1/R2) have only a relative +/- 0.423 % performance ..." on page 9. Some referred indices were not correct. For example, in the paragraph before Assumption 3 on page 6, "Assumption 1-->2 also requires that the convergence rates.... " The language problems are almost everywhere in the appendix. For example, "These environments covers wide applications includes tabular learning ....", "... for each environments." "... one experimental setup needs careful control... " The captions of Tables 2 and 3 ("... prioritized replAy..."). There are just way too many such problems all over the place. Finally, the authors stated on page "Detailed results for each environment is given in Appendix B." But the appendix does not contain any results.



**Summary Of The Paper:**

This submission presented a pessimistic model selection approach (PMS) for offline deep reinforcement learning (ORL) and tested the approach with performance comparison on six simulated environments. Under several specified conditions, the authors provided asymptotic analysis to show that PMS can lead to the best model with respect the worst performance of the derived policy.

**Summary Of The Review:**

The authors proposed the PMS to select ORL model based on the estimated worst performance to address potential overfitting issues in ORL. Both theoretical analysis and empirical performance have been conducted. The overall impression of the submission is that the authors should carefully check the writing and present more complete and comprehensive performance evaluation. Better presentation of the research results will be also appreciated.

---

> ### Author Response · Authors · 2021-11-22
> **Reply to Reviewer Sxiw**
>
> Thank you for providing suggestions and related discussions on our work. We summarized the related questions below.
>
> ***
>
> **Q1:** The authors may want to better explain some steps of the proofs in the appendix, for example, the last inequality for the proof of corollary 1 on page 17.
>
> **A1:** Following the reviewer’s suggestion, we went through the proofs and improved the clarity of the proofs, especially for the mentioned parts.
>
> ***
>
> **Q2:** Regarding the empirical evaluation of PMS, it appears that the theoretical analyses were mostly based on several references, including Su et al. (2020).
>
> **A2:** We believe there could be some misunderstandings about our contributions, and we would like to use this opportunity to clarify them. In terms of empirical evaluations, our numerical experiments and environments are comprehensive and diverse when compared to related works.  In terms of theoretical analysis, we remark that this is the first paper that considers a pessimistic approach to perform model selection in offline reinforcement learning. Therefore the proposed algorithm is new, and the corresponding theoretical analysis needs to be developed independently, while we admit that some part of our results will lead to similar findings compared to some references. In addition, while the proof idea in Corollary 2 is similar to Su et al. (2020) (the original idea is indeed from Lepski’s principle), our focus is on asymptotic property of our estimated value function for model selection in policy optimization instead of the finite sample bound for the off-policy evaluation developed in Su et al. (2020), which is a substantial difference in terms of theoretical contributions and the resulting algorithm.
>
> ***
>
> **Q3**: If it is the optimal policy, shouldn't the corresponding regret be zero? Also by simply looking at the tables in the appendix, it is not really clear how the initial models were selected in different experiments. More comprehensive performance evaluation should be provided.
>
>
> **A3:** We want to note that the optimal regret in our experiments is not zero since we can only use data to obtain $\hat \pi_{l}$ instead of $\pi_{l}$ for each model. We have also provided top-k results in our Appendix Figure 6 to 17, where the numerical results are close to zero. (e.g., <0.05)
>
> ***
>
> **Q4:** The submission may have to be checked carefully.
>
> **A4:** Thank the reviewer for pointing out the issue. We have fixed these typos and improved the clarity.
>
> ***
>
> **Q5:** The presentation needs to be improved.
>
> **A5** We thank the reviewer for the careful reading and suggestions. We have revised our paper accordingly.

---

> > ### Comment · Reviewer_Sxiw · 2021-11-27
> > **math notations**
> >
> > I truly appreciate the authors' efforts.
> >
> > I tried to read the revision quickly but found numerous typos, especially math notations. For example, in Eqn. (2), should $d^{\pi(s)}$ be $d^{\pi(s,a)}$? It is clearly not correct to have $s \in \mathcal{A}$ in the newly added definition above Eqn. (5) on page 5; and many others.
> >
> > The idea of PMS is interesting but I would suggest the authors to proofread their writing more carefully and also provide more comprehensive performance evaluation experiments if they can (weaknesses 2 in my original review).

---

> > > ### Author Response · Authors · 2021-11-27
> > > **Reply to Reviewer Sxiw**
> > >
> > > Thanks for your suggestion. We will fix the typos in the future version. In this revision, we have added more experiments to evaluate our methods. For example, we conducted a sensitivity analysis on different $\alpha$ and $O$. In the appendix C, we also reported the performance of our method in each separate environment (total six experiments). We additionally used top-K relative regret and top-K precision to further demonstrate the superior performance of the proposed method.

---

### Official Review · Reviewer_Wrcc · 2021-11-03

**Correctness:** 3
**Technical Novelty And Significance:** 2
**Empirical Novelty And Significance:** Not applicable
**Recommendation:** 3
**Confidence:** 3

**Main Review:**

Introduction mentions "AutoML" but I don't believe the main paper focuses on AutoML or even mentions it again. There seems to be a disconnect between this motivation and the content.

Saying their proposed PMS approach is “tuning free” is kind of overselling: you still have to pick the  $O$ and $\alpha$ hyperparameters, and additionally tuning the models architecture/training details for the value estimation procedure (probably FQI) that is used in each iteration to get $\hat{Q}$. Consider adjusting this claim; also compare to Zhang & Jiang 2021 [1] (this paper is very new so the authors are not expected to know/cite it, it is however important to illustrate what tuning free might mean).

Main results seem to have been aggregated over all 6 domains considered, however it is unclear whether the regret across domains are comparable i.e. on the same scale. It would be clearer to also provide per-domain results (the experiments section says these should be in Appendix B be but they are not).

What are $O$ and $\alpha$ set to for the experiments?

Since the method splits the data into $O$ parts, this will lead to higher variance estimates for each iteration compared to using the full dataset. Is this accounted for in the theoretical analysis?

[1] Siyuan Zhang, Nan Jiang. Towards Hyperparameter-free Policy Selection for Offline Reinforcement Learning. NeurIPS 2021. https://arxiv.org/abs/2110.14000

**Summary Of The Paper:**

This paper considers the model selection problem in offline RL and proposes a new procedure called pessimistic model selection (PMS) that estimates performance lower bound for each candidate policy. Theoretical results show that under certain assumptions PMS can identify the best candidate policy with high probability. The proposed approach is compared to three common OPE methods used in Tang & Wiens 2021 and shows favorable performance on 6 RL domains.

**Summary Of The Review:**

This paper makes an attempt at an important problem - offline RL model selection, however some experimental details are missing and the results seem to be too brief and not as informative as they could be. It also has significantly smaller page margins than the ICLR required format.

---

> ### Author Response · Authors · 2021-11-22
> **Reply to Reviewer Wrcc**
>
> Thank you for providing suggestions and related discussions on our work. We summarized the related questions below.
>
> ***
>
> **Q1:** Introduction mentions "AutoML" but I don't believe the main paper focuses on AutoML or even mentions it again. There seems to be a disconnect between this motivation and the content.
>
> **A1:**  We agree with the reviewer and we have removed AutoML in the revised version.
>
>
> ***
>
> **Q2:** Saying their proposed PMS approach is “tuning free” is kind of overselling ...
>
> **A2:**  Thanks for pointing out this. We have rephrased “tuning free” into “our approach essentially does not involve additional hyperparameter tuning except for two interpretable parameters” to prevent overselling. To make it more clear, our algorithm did not need to tune models for FQI, etc. Once the candidate models are determined to train Q-network, we will not change them anymore. There are only two parameters ($\alpha$ and $O$) which indeed need to be determined in advance but they are very interpretable to set up. $\alpha$ determines how pessimistic (robust) the selection is and $O$ is the trade-off for balancing computational costs and finite-sample accuracy. Indeed we can show that the variance of the estimated value function by our algorithm can achieve the semi-parametric efficiency bound, which is the best one can hope for. So in the asymptotic sense, the effect of $O$ is negligible. In the finite-sample setting, we believe the performance will be discounted by a factor $\sqrt{\frac{O}{O-1}}$. Therefore, if $O$ is large enough, $\sqrt{\frac{O}{O-1}}$ will have a mere effect on the performance.
>
> ***
>
> **Q3:**  Consider adjusting this claim; also compare to Zhang & Jiang 2021 [1] (this paper is very new so the authors are not expected to know/cite it, it is however important to illustrate what tuning free might mean).
>
> **A3:**  Thanks for pointing out this important literature. Unfortunately, we are not able to compare this method numerically within a short period since it is a very new paper (and also per the ICLR author guideline we are not required to compare any recent work published on or after June 5, 2021). Hence we decided to make some comments on this method, compared with ours. First of all, this is definitely a decent idea by using the class of piecewise constants created by other Q-functions to characterize the Bellman error. The algorithm proposed by Zhang & Jiang 2021 [1] seems to perform well in practice. One limitation of this method is that it requires state space to be finite. Without this assumption, Proposition 1 of Zhang & Jiang 2021 will not hold and mean squared of projected Bellman error may not be a good measure. Indeed it will be interesting to see if such a limitation could be addressed by using the $\ell_{\infty}$ norm in equation (1) or (2) of Zhang & Jiang 2021. In addition, there seems to be a goal-mismatch between choosing an optimal policy that maximizes the value function and minimizing the Bellmen error. In contrast,  our method does not suffer from the above two limitations. We have cited this work in the related work.
>
> ***
>
> **Q4:** Main results seem to have been aggregated over all 6 domains considered, however, it is unclear whether the regret across domains is comparable i.e. on the same scale. It would be clearer to also provide per-domain results (the experiments section says these should be in Appendix B be but they are not).
>
>
> **A4:** Thank you for pointing out this issue. We provide the per-domain results in Figure 6 to Figure 11 (Appendix section B) in the revised version. From our experimental results, PMS selection helps to attain a group of policies with lower regret in the six evaluated DRL environments v.s. six different offline reinforcement learning algorithms.
>
> ***
>
> **Q5:** Since the method splits the data into $O$ parts, this will lead to higher variance estimates for each iteration compared to using the full dataset. Is this accounted for in the theoretical analysis?
>
> **A5:** Our method will not lead to higher variance. The final output policy makes use of the full dataset to train, so it will not suffer from higher variance. The estimator for optimal value function can be shown to achieve semi-parametric efficiency. Therefore asymptotically speaking, the procedure will not suffer from efficiency loss. However, empirically, we believe the performance of estimated optimal policy may be discounted by  $(O-1)/O$. When $O$ is large, this seems negligible.
>
> ***
>
> **Q6:** It also has significantly smaller page margins than the ICLR required format.
>
> **A6:** We did not tweak the page margins. Could the reviewer point out where the format issue could be?
>
>
> ***
>
> [1] Siyuan Zhang, Nan Jiang. Towards Hyperparameter-free Policy Selection for Offline Reinforcement Learning. NeurIPS 2021. https://arxiv.org/abs/2110.14000

---

> > ### Comment · Reviewer_Wrcc · 2021-11-26
> > **Reply to authors' response**
> >
> > Dear authors,
> >
> > Thank you for answering my questions, and revising the introduction/discussion on AutoML, and providing the per-domain experimental results.
> >
> > Re page margin: there might have been a latex compile issue in your original submission. In your revised pdf this seems to have been fixed. You may disregard my earlier comment on this.
> >
> > Unfortunately not all my questions have been addressed. For example Re "What are $O$ and $\alpha$ set to for the experiments" While the authors added Appendix C.1 for sensitivity analyses of different $O$ and $\alpha$ values on one of the tasks, it is still unclear what values were used in the main experiments and what was the rationale for selecting them, especially because there seems to be non-negligible variability in results depending on $O$ and $\alpha$.
> >
> > In light of the remaining issues and other reviewers’ concerns, I am not adjusting my score.

---

> > > ### Author Response · Authors · 2021-11-27
> > > **Re: Reply to authors' response**
> > >
> > > Dear Reviewer  Wrcc,
> > >
> > > Thank you for your follow-up suggestion. We provide related feedback and clarification below.
> > >
> > > - Value of $\alpha$ and $O$
> > >
> > > The main focus of our work is on the theoretical approaches and findings. For our experiments, we fixed the $\alpha$ to $0.01$ and $O$ to $20$ for all environments. Different values of $O$ and $\alpha$ do not significantly affect the performance of model selection empirically (see Figures 18 and 19 and the discussion in Appendix C.1). Even in a smaller case of $\alpha$ value, PMS still attain the best results as a new model selection algorithm.
> > >
> > >
> > > ***
> > >
> > > - Margin
> > >
> > > Thank you for the follow-up clarification. From the latex log history, we **did not change** any space arguments. We are glad this concern is resolved.

---

### Decision · Program_Chairs · 2022-01-20

**Decision:**

Reject

**Comment:**

The paper proposes a new approach called pessimistic model selection (PMS)  for model selection in offline RL and tests it in 6 different environments. Under certain assumptions this allows theoretical results that the best model is recovered with high probability.

Several points were raised by the reviewers and maintained after the rebuttal:
- Theoretical results were considered weak as they only hold asymptotically.
- Experimental results limited (potentially different regret scales, no sufficient comparison to other baselines).
- Exposition of the paper that needs to be improved.

Given the strong consensus among the reviewer I recommend rejecting this paper.